# Roving methyltransferases generate a mosaic epigenetic landscape and influence evolution in *Bacteroides fragilis* group

Michael J. Tisza [1,4,7], Derek D. N. Smith[1,5,7], Andrew E. Clark[2,6], Jung-Ho Youn[2], NISC Comparative Sequencing Program*, Pavel P. Khil[1,2] & John P. Dekker [1,2] ✉

Three types of DNA methyl modifications have been detected in bacterial genomes, and mechanistic studies have demonstrated roles for DNA methylation in physiological functions ranging from phage defense to transcriptional control of virulence and host-pathogen interactions. Despite the ubiquity of methyltransferases and the immense variety of possible methylation patterns, epigenomic diversity remains unexplored for most bacterial species. Members of the *Bacteroides fragilis* group (BFG) reside in the human gastrointestinal tract as key players in symbiotic communities but also can establish anaerobic infections that are increasingly multi-drug resistant. In this work, we utilize long-read sequencing technologies to perform pangenomic ($n = 383$) and panepigenomic ($n = 268$) analysis of clinical BFG isolates cultured from infections seen at the NIH Clinical Center over four decades. Our analysis reveals that single BFG species harbor hundreds of DNA methylation motifs, with most individual motif combinations occurring uniquely in single isolates, implying immense unsampled methylation diversity within BFG epigenomes. Mining of BFG genomes identified more than 6000 methyltransferase genes, approximately 1000 of which were associated with intact prophages. Network analysis revealed substantial gene flow among disparate phage genomes, implying a role for genetic exchange between BFG phages as one of the ultimate sources driving BFG epigenome diversity.

Methylation of genomic DNA has been detected in all three domains of cellular life as well as in viruses[1–3]. Eukaryotic genomes display dynamic methylation of cytosine at the C5 position (5mC) within certain CpG (5′-CG-3′) contexts, and regulation of this CpG methylation at specific sites affects transcription[4], genome repair dynamics, and genome compaction[5]. In contrast, bacteria display motif-specific DNA methylation (e.g., 5′-CC-6mA-TGG-3′) where nearly all instances of a given motif may be methylated[6]. Similar to eukaryotic genomes, 5mC modifications are common; however, bacterial genomes display additional methylation at the N4 position of cytosines (4mC) and, most commonly, the N6 position of adenines (6mA)[6]. Bacterial DNA methylation is conducted by DNA methyltransferases, some of which appear to be present and active in all strains of a given species (e.g., Dam, which modifies GATC in *Escherichia coli*), whereas other DNA methyltransferases and the genes

[1]Bacterial Pathogenesis and Antimicrobial Resistance Unit, LCIM, NIAID, NIH, Bethesda, MD, USA. [2]National Institutes of Health Clinical Center, NIH, Bethesda, MD, USA. [4]Present address: The Alkek Center for Metagenomics and Microbiome Research, Department of Molecular Virology and Microbiol, Baylor College of Medicine, Houston, TX, USA. [5]Present address: Environment and Climate Change Canada, Ecotoxicology and Wildlife Health Division, Wildlife Toxicology Research Section, Ottawa, ON, Canada. [6]Present address: Department of Pathology, University of Texas Southwestern Medical Center, Dallas, TX, USA. [7]These authors contributed equally: Michael J. Tisza, Derek D. N. Smith. *A list of authors and their affiliations appears at the end of the paper. ✉e-mail: john.dekker@nih.gov

that encode them are transiently gained and lost over time and are not essential for viability in culture[7]. Classically, bacterial DNA methylation has been understood as primarily a byproduct of anti-phage defense based on restriction-modification systems[8]. However, other physiological consequences of maintaining methylated DNA, often at thousands of loci, have now become clear. Studies have demonstrated roles for bacterial DNA methylation in the regulation of transcriptional activity controlling virulence phenotypes[9–11] and other physiologic programs[12,13], genome stability[14,15], and affecting mutation frequency within methylated motifs[16,17], similar to observations in eukaryotic systems.

Bacteria in the *Bacteroides fragilis* group (BFG) represent more than a dozen species in the *Bacteroides*, *Parabacteroides*, and the recently introduced *Phocaeicola* genera[18]. These abundant symbiotes can readily be found living anaerobically in human gastrointestinal tracts and have been implicated in many important metabolic and immune functions[19–21]. They are also among the most commonly recovered bacteria from extra-intestinal anaerobic infections and are increasingly resistant to many antibiotics, including cephalosporins and carbapenems[22,23]. Their broad phenotypic purview is enabled in part by phase variation, an array of polysaccharide utilization loci, and their use of invertible promoters[24,25].

In this work, clinical isolates from an historical collection of BFG spanning four decades were studied using a combination of short and long-read genomic sequencing, methylome analysis, and antimicrobial susceptibility phenotyping. The comprehensive scope of the methylome analysis performed in this study, in combination with contiguous long-read assemblies, revealed an epigenetic landscape in clinical BFG isolates of immense and previously unappreciated diversity. Hundreds of DNA methylation motifs containing 5mC, 4mC, and 6mA were identified across the genomes, with nearly all motif combinations observed only in single isolates. Some DNA methylation motifs were strongly enriched within particular lineages within a species, and evidence of genome-wide depletion of these motif sequences was frequently observed in these same lineages, suggesting selection and pointing to DNA methylation as a driver of genome evolution in the BFG.

## Results

### In-depth characterization of an historical collection of 383 BFG clinical isolates

More than 600 clinical BFG isolates cultured during the course of routine care of patients in the NIH Clinical Center in Bethesda, MD, USA were collected and cryogenically stored between 1973 and 2018. A set of 383 isolates was selected from this collection to represent a range of dates, species, and antimicrobial resistance profiles. Isolate genomes were sequenced with long-read nanopore sequencing (*n* = 383), and a representative subset (*n* = 13) received additional PacBio SMRT sequencing (Supplementary Data 1). De novo assembly of genomes was performed, and in 68.1% (261/383) of isolates, chromosomes were assembled as a single, circular contig (Supplementary Fig. 1A), ranging in length from 3.9 to 7.2 megabases. Evaluation of assembly quality indicated that long repetitive regions could be resolved in these assemblies. For example, some isolates contained more than ten tandem and non-tandem copies of Tn4555, a 12 kilobase (kb) transposon carrying *cfxA*, a beta lactamase gene[26], and the long read approach allowed resolution of copy number and genomic locations of these repeats (Supplementary Fig. 1B). Further, analysis of circularized genomes showed that between 3 and 7 rRNA operons (>5 kb each) could be detected in assemblies (Supplementary Fig. 1C). The numbers of identified rRNA operons per circular chromosome corresponded to the expected values for the species in almost all cases based on data derived from the Ribosomal RNA Database[27].

Taxonomy of each isolate was investigated with two methods. First, the Bruker Biotyper[28,29] was used to analyze bacterial lysates by MALDI-TOF mass spectrometry. Then, the GTDB-Tk[30] was employed

on genome sequences to place each in a species-level bin (Fig. 1). These methods were largely congruent, agreeing on 360/383 isolate genomes (94.0%), though different numbers of final species designations were reported, with 15 identified by MALDI-TOF and 21 by GTDB-Tk. These discrepancies are in part explained by the fact that the GTDB-Tk uses newer taxonomy structure that has split some relevant species/ genera. On the basis of Bruker Biotyper identification, *Bacteroides fragilis sensu stricto* was the most common species in the set of isolates, contributing 135 unique species-level assignments, followed by *Bacteroides thetaiotaomicron* (*n* = 80), *Bacteroides ovatus* (*n* = 51), and *Bacteroides vulgatus* (*n* = 32) (see Methods). The genetic diversity of this dataset was visualized by the pairwise nucleotide similarity distances between all isolate genomes (Supplementary Fig. 2), showing clear species-level clustering. It should be noted that the name *Phocaeicola vulgatus* has recently been proposed and accepted for *Bacteroides vulgatus* (*B. vulgatus*)[29]. The name *B. vulgatus* is retained throughout this manuscript for consistency with most of the existing literature.

Antimicrobial susceptibility testing was performed for seven antibiotics (ampicillin, ampicillin/sulbactam, piperacillin/tazobactam, meropenem, metronidazole, moxifloxacin, clindamycin, and tetracycline) on 324 sequenced isolates by the reference agar dilution method (Fig. 1 and Supplementary Data 2). This testing demonstrated heterogeneous and complex resistance patterns amongst isolates of each species with the isolates from the most recent decade exhibiting similar resistance patterns to other published work[23,29]. Consistent with prior studies, we found that resistance to several antibiotics, including piperacillin-tazobactam and meropenem appears to have increased in certain species such as *B. fragilis* and *B. ovatus* from the 1980s to the 2010s (Supplementary Fig 3A). This was supported to some extent by a concomitant increase in certain antimicrobial resistance genes over the same period (Supplementary Fig. 3B).

### Genomic analysis demonstrates open pangenomes and substantial movement of mobile genetic elements within and between BFG species

Pangenome analysis[31] of eight species from the current study supplemented with additional GenBank reference genomes revealed that the accessory (cloud and shell) gene families in each species varied from 29.0% (*Bacteroides faecis*) to 42.2% (*B. ovatus*) of total gene content (Fig. 2a). Further, rarefaction analysis (Fig. 2b), and Heap's law estimates (Supplementary Table 1) demonstrated that the pangenome of each species remained open with some species containing >20,000 sampled genes within the dataset, implying that an immense number of additional gene families await discovery within the BFG. This pangenome openness is largely consistent with gut-derived *Bacteroides* metagenome assembled genomes[32].

To understand the flow of genes and mobile genetic elements across genomes and species, 31,436 accessory regions (DNA sequences >3 kb in length encoding only accessory genes, Supplementary Data 3) were extracted from 414 genomes representing 13 species for which three or more genomes were available (378 genomes from this study and 36 genomes from NCBI)[33]. Comparison of each accessory region sequence with all others in this set demonstrated that >10% of such regions were shared between species, suggesting horizontal transfer (Fig. 2c). Each accessory region was probed for a variety of features, and it was found that phage, phage defense systems, DNA methyltransferases, conjugative machinery, episomes/plasmids, and antimicrobial resistance (AMR) genes were all more common in accessory regions detected in three or more species (Fig. 2c). For instance, accessory regions encoding the tetracycline resistance gene *tet(Q)* and/or a cassette with genes *tet(X)1*, *tet(X)2*, and the aminoglycoside modifying enzyme *aadS* were detected in 12 of 13 species analyzed (Fig. 2 and Supplementary Fig. 4), likely confirming a history of selective pressure from tetracycline and aminoglycoside compounds.

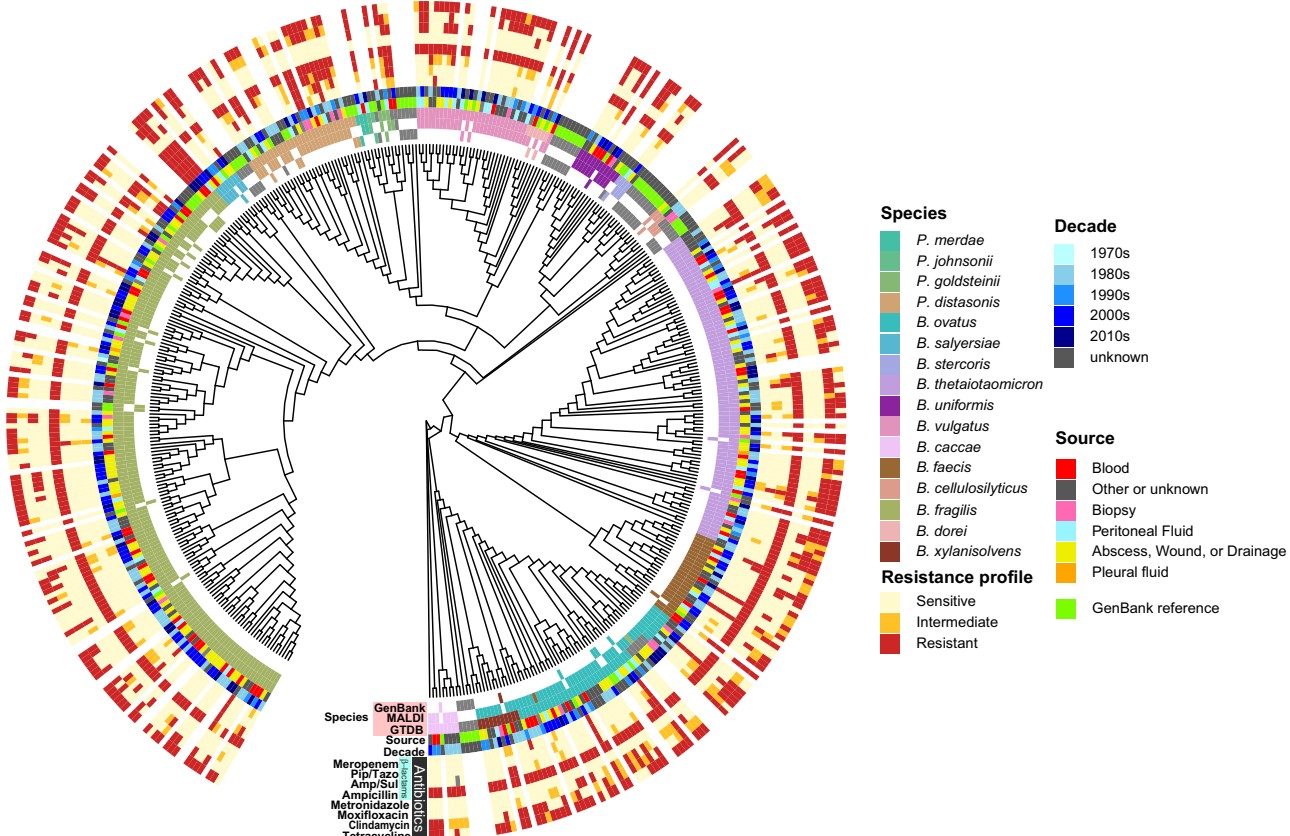

**Fig. 1 | Four-decade collection of clinical BFG isolates from the NIH Clinical Center.** MLST marker gene cladogram of BFG genomes sequenced in this study supplemented with Genbank reference genomes ($n = 462$ total). Taxonomy assignments were defined proteomically with MALDI-TOF mass spectrometry (Bruker Biotyper) and genomically with GTDB-Tk. "Source" and "Decade" data were extracted from clinical laboratory metadata records.

Many pathogenic bacteria of medical importance, including *Enterobacterales* and related Gammaproteobacteria, carry a large proportion of AMR genes extra-chromosomally on plasmids[34]. The dataset analyzed in this work yielded 575 complete circular plasmids or episomes across the 383 sequenced isolate genomes (Supplementary Data 4), belonging to 85 clusters of >95% average nucleotide identity (see Methods) (Supplementary Fig. 5A). The majority of circular contigs (550 out 575; 95.7%) had recognizable plasmid genes such as replicases or relaxases (see Methods), and a proportion of the remainder may represent replicative intermediates of transposons, but this was not analyzed further. Despite the ubiquity of both plasmids/episomes and AMR genes in the sequencing data, we found that most of these AMR genes were not located on plasmids/episomes (53 out of 1911 AMR genes were located within circular plasmid/episome contigs) among BFG species. The overwhelming majority (>97.2%) of AMR genes appeared to be located within chromosomes, and many were associated with integrative elements[23]. Many of the AMR genes encoded by plasmids/episomes also appeared associated with integration of integrative elements into plasmid backbones, consistent with possible shuttling of AMR genes between chromosomes and plasmids/episomes (Supplementary Fig. 5B).

## DNA methyltransferases are remarkably diverse and abundant in the accessory genome

It is hypothesized that DNA methyltransferases may function as a class of global regulators in many bacterial species[7]. Methyltransferases usually modify DNA at short motifs present at thousands of sites scattered broadly across intragenic regions and gene bodies of bacterial genomes, and thus the expression of a single methyltransferase gene may in turn control global methylation states. Methylation at both intra- and intergenic sites is known to affect transcriptional programs and tune bacterial phenotypes[9–13]. A significant proportion of bacterial methyltransferase genes in turn have been observed in association with mobile genetic elements, particularly within accessory regions of bacterial genomes[35,36]. To facilitate identification of methyltransferases in BFG genomic data, we built on previous hidden Markov model approaches[37] to develop a publicly accessible tool, DNA Methylase Finder, to detect and annotate DNA methyltransferase genes and the gene neighborhoods to which they belong (https://github.com/mtisza1/DNA_methylase_finder). A sensitivity of 100% and false positive rate of up to 5.4% were observed in a benchmarking study of this tool with a REBASE dataset of annotated methylation motifs (see Methods).

Using DNA Methylase Finder, 6011 DNA methyltransferase genes were detected in 462 BFG genomes (genomes from this study supplemented with additional BFG genomes downloaded from GenBank) (Supplementary Data 5). These genes were binned into 536 families (Supplementary Data 6) (see Methods), representing all known types (Type I, Type II, Type IIG, Type III, as well as DNA methyltransferases that could not be classified, labeled as "unknown"). Between two and 38 DNA methyltransferase genes were detected in each genome analyzed, and methyltransferase gene families could be found in the persistent, shell, and cloud partitions of the pangenomes of these species, showing a wide spectrum of mobility (Fig. 3). Of the 5480 DNA methyltransferase genes belonging to the 15 analyzed species in Fig. 3, 720 (13.1%) were in the core partition, 2385 (43.5%) were in the shell partition, and 2375 (43.3%) were in the cloud partition.

The number of methyltransferases we identified with this method may be greater than would be anticipated based on previous reports. We expect that some fraction of the identified putative methyltransferases

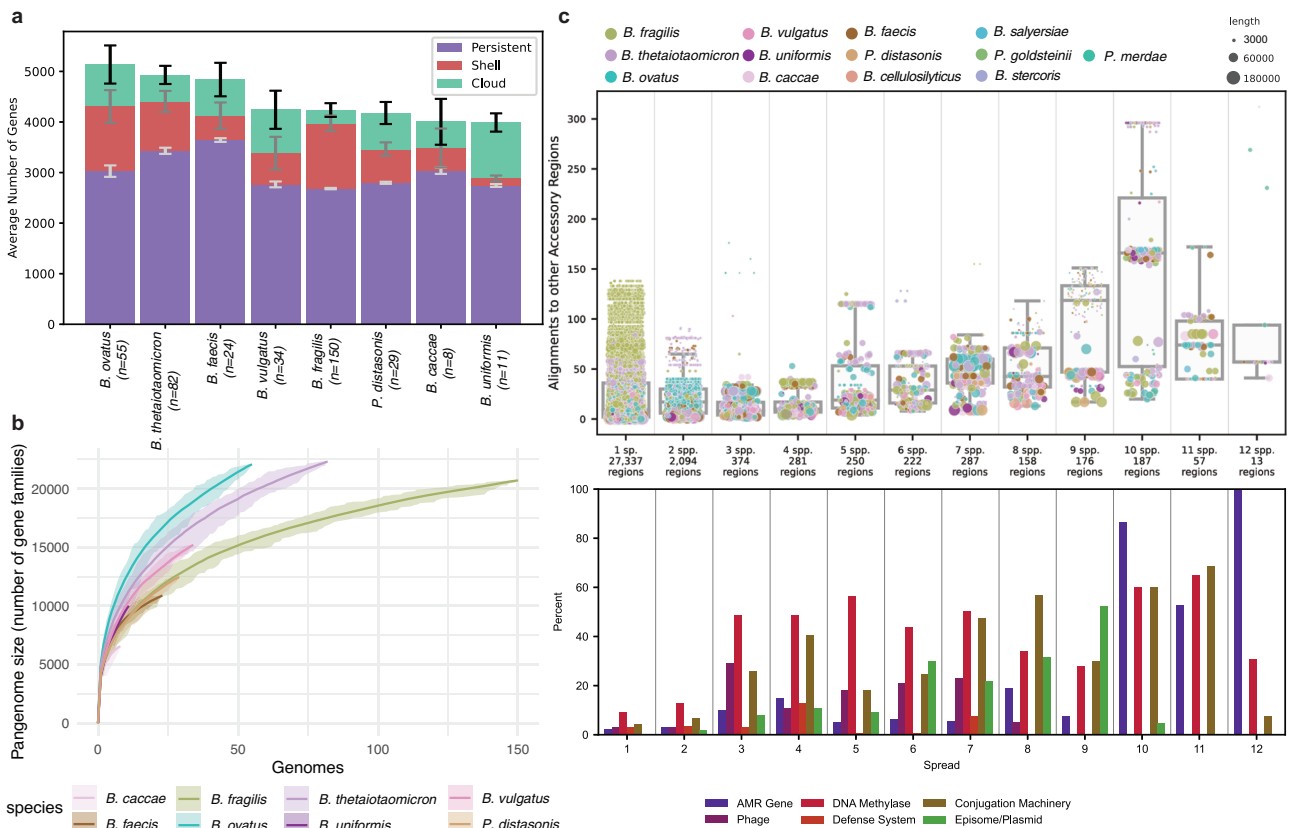

**Fig. 2 | Genomic analysis demonstrates open pangenomes and substantial movement of mobile genetic elements within and between species. a** Stacked barplots quantifying average numbers of persistent, shell, and cloud genes across eight species. **b** Pangenome analysis for a subset of BFG species. Rarefaction curves indicate open pangenomes over the sequenced set, with the three largest pangenomes demonstrating greater than 20,000 genes each. **c** Analysis of accessory region and mobile genetic element content. Top panel shows species-level spread bins for more than 33,000 accessory regions/mobile genetic elements. "Species" indicates number of species sharing the indicated number of accessory regions or mobile genetic elements. Paired bottom panel barchart indicates annotated features of the accessory regions as a percentage of each paired spread-level bin in the upper panel.

are likely inactive, and additionally, the method demonstrated a false positive discovery rate of up to 5.4% when measured against the REBASE database, so some small percentage may be false positive identifications. However, we also noted that many genomes contained more than one methyltransferase of near identical sequence, in many cases in association with mobile genetic element context. Thus, the large number may be accounted for, in part, by duplications due to transposon insertions. It is very possible that studies based on short read sequencing may underestimate the number of such methyltransferase duplicates due to collapse during assembly. The 38 putative methyltransferases encoded by isolate BFG-632 (Supplementary Data 7), a number of which appeared to be duplicate insertions, were queried with BlastP and 37/38 returned hits with >90% amino acid identity to previously identified DNA methyltransferases. Interestingly, in the course of our analysis, we observed that within each species, there is a positive correlation between genome size and number of putative DNA methyltransferases (Supplementary Fig. 6). BFG-632 is the longest genome in the entire collection, consistent with the greatest number of methyltransferases.

Additional annotation of genes upstream and downstream of the identified putative DNA methyltransferases demonstrated that specificity subunits are detected almost exclusively near putative Type I DNA methyltransferases (Supplementary Fig. 7A). Additionally, restriction endonucleases were detected in the neighborhood of 100% of putative Type III DNA methyltransferases, and most Type II DNA methyltransferases are apparent orphans, without vicinal restriction endonucleases identified (Supplementary Fig. 7B). These additional features increase the confidence in many of these methyltransferase identifications.

## Network analysis reveals substantial methyltransferase gene flow among disparate phages

Annotation of gene neighborhoods of DNA methyltransferases above indicated that DNA methyltransferase genes were often found in proximity to phage-related genes. To examine this relationship in greater detail, putative prophage regions of each genome were extracted by scanning accessory regions with Cenote-Taker 2[38] and CheckV[39], revealing 1255 candidate prophage regions, most of which were predicted to be complete genomes (Supplementary Data 8–10). The majority ($n = 824$) of these prophages encoded at least one DNA methyltransferase gene, accounting for 1089 of the 6011 DNA methyltransferase genes in the genome set. The 1255 putative prophages could be clustered into 411 virus Operational Taxonomic Units (vOTUs) (Supplementary Data 9) (see Methods). Notably, there was substantial diversity of methyltransferase gene content within individual vOTUs, in combination with wide dispersion of single methyltransferase gene families across disparate phage genomes (Fig. 4 and Supplementary Fig. 8). Overall, this suggests not only that there is substantial methyltransferase gene flow among BFG phage genomes, but also that disparate BFG phage genomes may serve as important sources of genetic diversity for one another, possibly allowing recipients to subvert restriction-modification systems[35].

## BFG methylation motifs tile a vast combinatoric space with open panepigenomes

Oxford Nanopore sequencing technology has been used to identify 6mA, 4mC, and 5mC modifications with recently developed methods.

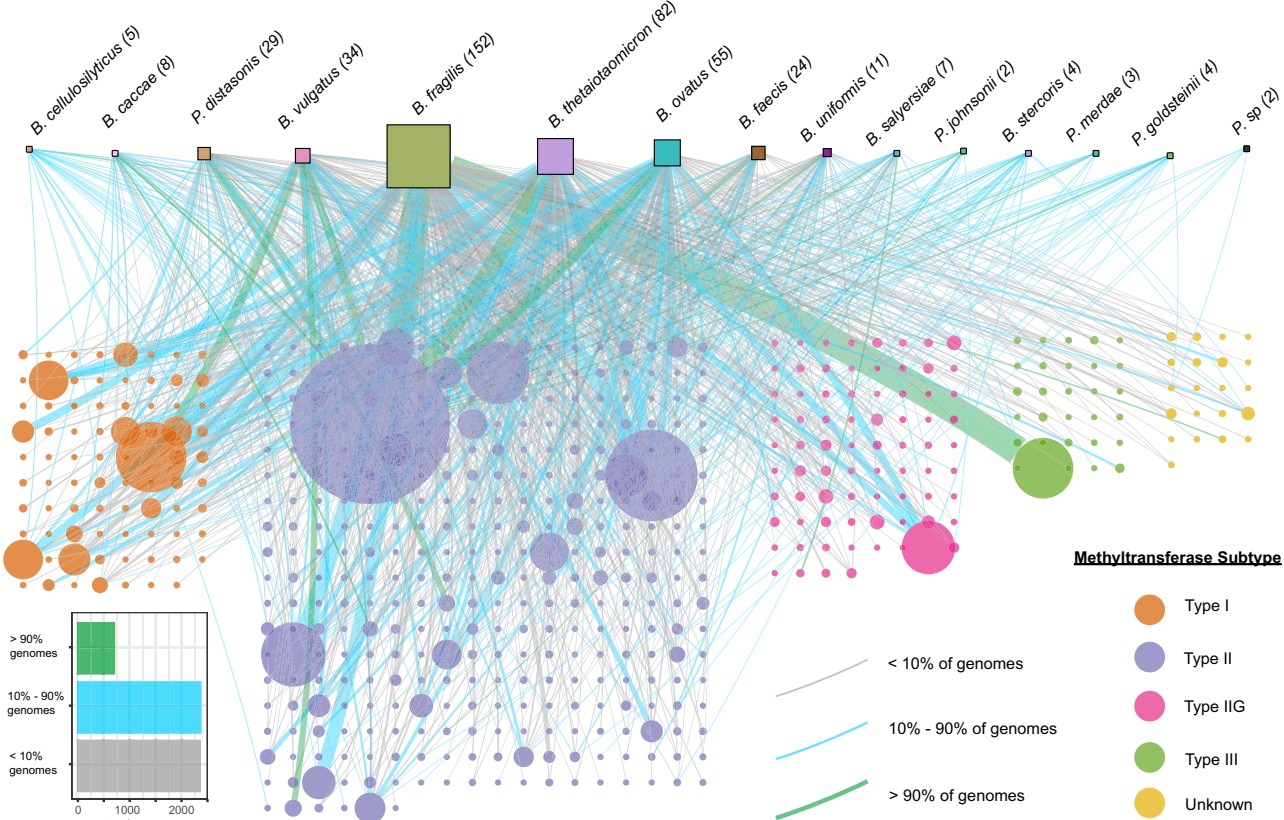

**Fig. 3 | DNA methyltransferases are remarkably diverse and abundant in the accessory genomes of BFG species.** Host species are represented across the top row as squares (area proportional to number of genomes analyzed). DNA methyltransferase gene families (80% AAI, 80% AF) are represented in rectangular grids below as filled circles (with area proportional to number of genes in the family) with colors indicated in the key. Edges connect species with DNA methyltransferase gene families that are encoded by one or more genomes within the species. Location of the given methyltransferase gene family in the core, shell or cloud genome in is indicated by edge color, and edge thickness indicates the number of times the gene family is encoded in the genome of the species. In this analysis, 'Core' was defined as presence in >90% of genomes in a species, 'Shell' was defined as presence in >10% and ≤90% genomes in a species, and 'Cloud' was defined as presence in <10% of genomes in a species.

Nanodisco is a powerful approach for methylation pattern detection that works by comparing raw current-level nanopore sequencing traces for native methylated genomic DNA to prepared unmodified DNA[40]. To benchmark methylation calls made by Nanopore and Nanodisco for this dataset against another method, PacBio and Nanodisco methylation motif identification were performed for a subset of six isolates representing six species for which sequencing data for both methods were obtained (Methods). This comparison revealed concordance of results for 6mA and 4mC. PacBio SMRT sequencing identified 29/33 6mA motif calls and 2/2 4mC motif calls made by Nanodisco. Two 5mC calls made by Nanodisco were not identified by PacBio sequencing, which is consistent with observed lower sensitivity of the PacBio approach for 5mC (Supplementary Table 2).

The Nanodisco method was then applied to 268 genomes from the BFG collection spanning five species, with manual curation of methylation motifs (as described in Methods and Supplementary Figs. 9–10). A total of 639 distinct methylation motifs were detected by de novo discovery (Fig. 5 and Supplementary Data 11). Remarkably, the number of distinct methylation motifs appears far from saturation in the analyzed dataset based on rarefaction curves (Fig. 5a) and Heap's law estimates (Supplementary Table 3), suggesting an immense number of total motifs used by the BFG. In addition to this diversity of individual methylation motifs seen in this sample set, most combinations of motifs were unique, present only in single isolates, generating an additional layer of combinatorial diversity, and suggesting a vast number of motif combinations within

BFG that have not yet been sampled (Supplementary Fig. 11). Though most motifs were detected in only a single species, two motifs (CTGCAG, and GATC) were detected in at least one isolate from all five analyzed species. A study of *Bifidobacterium breve* isolates using both PacBio and bisulfite sequencing[41], and another study looking at *Clostrioides difficile* using only PacBio SMRT sequencing (lower sensitivity for 5mC without method modification[9]), showed greater saturation of the panepigenome in those taxa (data from these studies plotted in Fig. 5a). It is possible that a less diverse sampling set or lower sensitivity for detecting motifs could generate an apparent saturation at lower genome coverage. These results nevertheless suggest that BFG species may contain a greater diversity of methylation motifs and motif combinations than other gastrointestinal anaerobes.

For each species in the analysis (*B. fragilis*, *B. thetaiotaomicron*, *B. ovatus*, *B. vulgatus*, and *P. distasonis*), the presence or absence of each DNA methylation motif was investigated in relation to a phylogenetic tree of marker genes within the species (Fig. 5b–f). As noted above, the majority of methylated motifs were present only in single isolates in the set. A small number of DNA methylation motifs were methylated in all isolates of a species (e.g., CTCAT in *B. fragilis* or CGCG, CCAGG, and CCTGG in *P. distasonis*). Some motifs were methylated mostly or entirely within a sub-species lineage (e.g., GATC in *B. ovatus*), while other motifs appeared to be distributed irrespective of phylogeny (e.g., CCWGG in *B. thetaiotaomicron*). At least one methylated motif was detected in all genomes except within two *B. thetaiotaomicron* isolates and one *B. vulgatus* isolate.

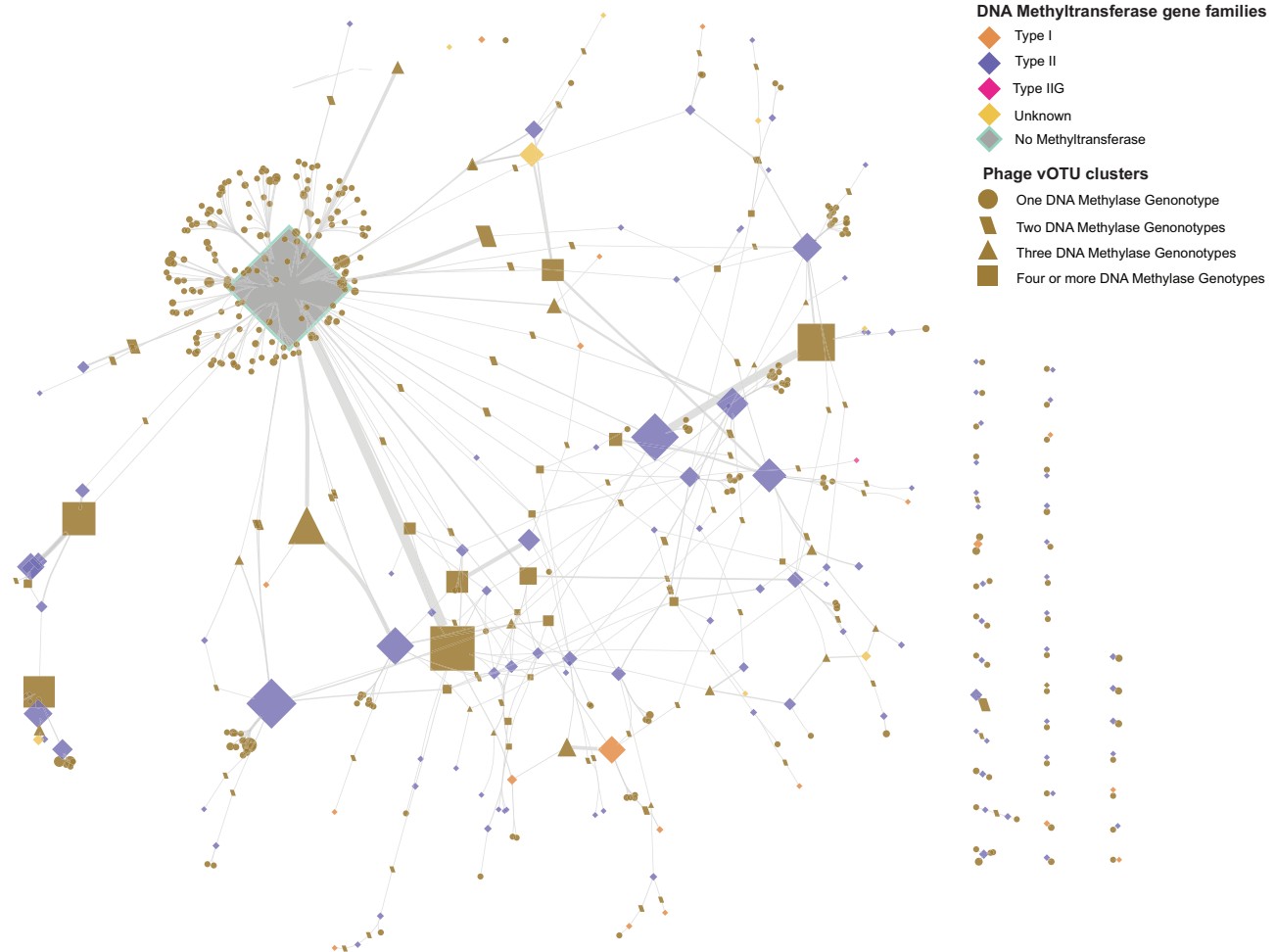

**Fig. 4 | Network analysis reveals substantial methyltransferase gene flow among disparate phage genomes.** Network graph of phage viral operational taxonomic units (vOTU) clusters and DNA methyltransferase gene families (80% AAI, 80% AF), indicated by nodes of different shapes as defined in the legend. Shape size is proportional to the number of phage genomes within a given vOTU cluster or methyltransferase genes within a gene family. Edges connect methyltransferase gene families and vOTU clusters containing prophage genomes that encode a methyltransferase gene from that methyltransferase gene family. Edge thickness is proportional to the number of genomes that encode the corresponding gene family.

## AMR genes and promoters contain abundant DNA methylation motifs

Transcriptional regulation of AMR genes is known to play an important role in the expression of resistance phenotypes in many different species, and recent work has demonstrated that AMR gene expression and resistance phenotypes can be specifically regulated by methylation[10,42]. We thus we searched for DNA methylation motifs in the gene body and the promoter of AMR genes in BFG that may influence transcription. To investigate the frequency and distribution of such motifs, AMR genes and their upstream regions (200 nucleotides) from *B. fragilis*, *B. thetaiotaomicron*, *B. ovatus*, *B. vulgatus*, and *P. distasonis* genomes were extracted and dereplicated at 99% nucleotide identity (see Methods). These AMR gene regions were then profiled for the presence of motifs that were found in at least one genome from the corresponding species (Supplementary Fig. 12A–E). Remarkably, every profiled AMR region had multiple DNA methylation motifs in the gene body and upstream/promoter region detected in at least some isolates.

To ask the further question of whether motif density in the AMR gene bodies differs from that of the rest of the genome, we performed an analysis of methylation in AMR genes relative to non-AMR genes in the five principal species for which there were sufficient numbers of isolates. A direct correlation between motif content and GC content became apparent in this analysis. Because of this

correlation, motif density was analyzed as a function of GC content for AMR genes vs. non-AMR genes. This analysis found no aggregate systematic differences between motif density (adjusted for GC content) in AMR genes vs. non-AMR genes (Supplementary Fig. 13). Whether methylation in any of these AMR genes or associated promoter regions have consequences for antimicrobial resistance warrants further investigation.

## Lineage-specific DNA methylation motifs show evidence of depletion

A subset of DNA methylation motifs displayed a strong phylogenetic signal, present in most or all closely related genomes, but seen rarely or not at all in more distantly related genomes of the same species. This enrichment was interpreted to indicate that genomic positions with this motif have been methylated in these lineages since the last common ancestor. If there is a negative fitness cost associated with tolerating these modifications at some loci in the genome following the introduction of a methyltransferase, one may expect to see a depletion of these motifs in genomes of the lineage containing the methylase due to selection. Additionally, methylation in certain contexts has been linked to hypermutation of the modified base resulting in programmed motif self-destruction[16].

We identified 14 lineage-specific motifs (see Methods), and 6 of these 14 (42.9%) appeared to be significantly depleted in the lineage

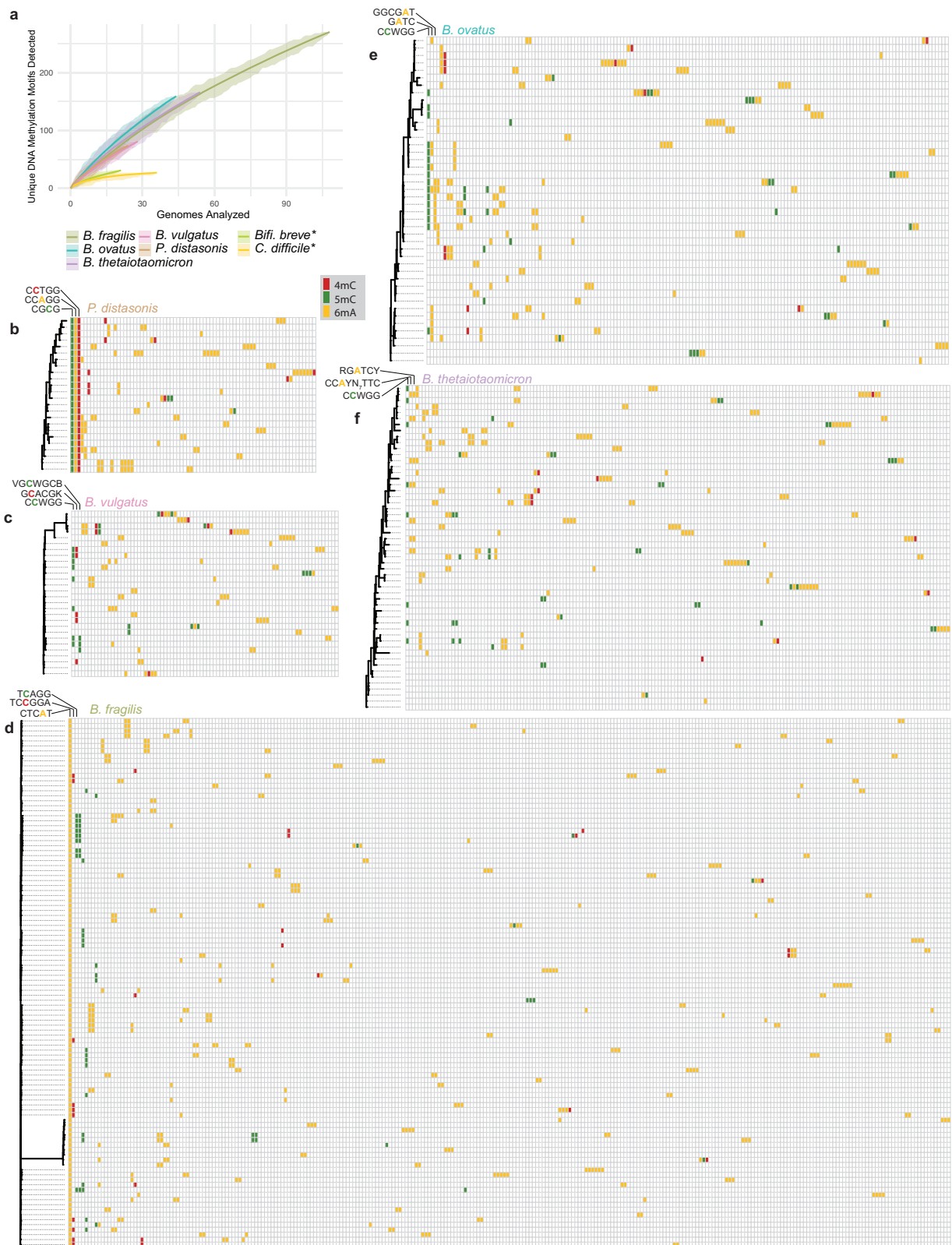

**Fig. 5 | BFG methylation motifs tile a vast combinatoric space with open panepigenomes over the sequenced isolate set. a** Rarefaction curves of DNA methylation motifs detected in genomes of BFG species in this study and comparison with *C. difficile* and *B. breve* species from external studies (denoted with '*'; data[9,41]). BFG rarefaction curves indicate open panepigenomes over the sequenced isolate set. **b**–**f** Heatmaps of detected DNA methylation motifs in *Parabacteroides distasonis, B. vulgatus, B. fragilis (sensu stricto), B. ovatus, and B. thetaiotaomicron*

isolates. Rows indicate individual isolates with corresponding marginal MLST marker gene phylogenies. Columns indicate distinct methylation motifs. The three most prevalent motifs in each set are labeled, and labels are omitted for the rest. Cells are colored when a given motif is present in the corresponding isolate and colors indicate the class of base modification as indicated in the legend. Sequences of all motifs can be found in Supplementary Data 10.

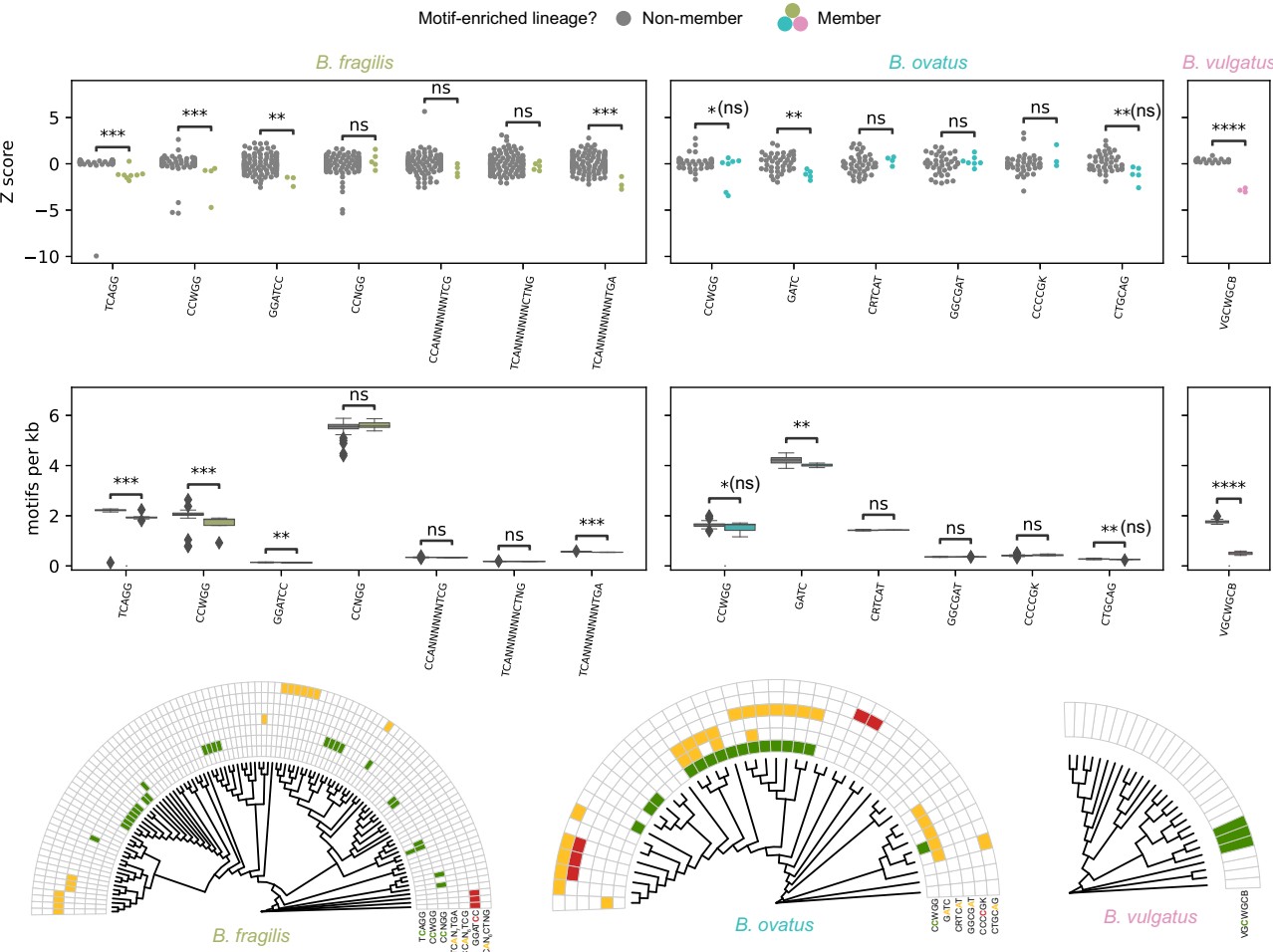

**Fig. 6 | DNA methylation motifs demonstrate genome-wide depletion in lineage-specific context.** (Top) A swarm plot of Z scores of density (motifs/kilobase) of each motif across genomes. Two-sided T-tests were performed for each target motif along with its control motifs and Benjamini–Hochberg testing performed separately with FDR < 1%. Unadjusted *p* values are reported as follows: *$p \leq$ 0.05, **$p \leq$ 0.01, ***$p \leq$ =1e-3, ****$p \leq$ =1e-4; "ns" indicates that the reported *p* value was determined to be non-significant after testing with Benjamini-Hochberg at FDR = 1%. (Middle) Data displayed as motifs per kilobase. Boxes show data quartiles and whiskers show 1.5X IQR with diamonds representing outliers. (Bottom) MLST marker gene cladograms by species using all methylome-analyzed genomes with heatmap of lineage-specific DNA methylation motifs. (All) *B. fragilis* genomes, *n* = 108. *B. ovatus* genomes, *n* = 44. *B. vulgatus* genomes, *n* = 28.

genomes after multiple test correction while none were enriched (Fig. 6). For comparison, motif density for 16 to 58 control motifs of the same length and base composition obtained by permutation (e.g., GATC control motifs include AGTC, ATCG, and CTAG) was calculated for each lineage-specific motif. If more than 58 possible control motifs existed, 50 were chosen by random shuffling. Between 0% and 13.7% of permuted motifs were depleted (average of 3.8%) and between 0% and 37.5% were enriched (average of 5.1%) (Fig. 6, Supplementary Figs. 14–21, Supplementary Table 4). Notably, of the six lineage-specific motifs that appeared to be depleted at a significant level, five motifs were palindromic in such a way that each locus with these motifs had a methylated base on both strands. The non-palindromic motif that was depleted (TCAGG/CCTGA) is a type IIS motif, in which the motifs are reverse complements of each other, and thus both strands of DNA at these loci are methylated as well. Whether selection acts differentially on motifs that are methylated on both strands is a further question that we are unable to evaluate definitively in this dataset. We did not observe an enrichment of motifs that would result from transitions or transversions at the modified site for 5mC and 4mC modifications that could not account for the magnitude of the depletions (Supplementary Fig. 21), suggesting that hypermutation alone cannot explain the findings[16,17].

## Discussion

The global methylome analysis performed in this work, in combination with contiguous long-read assemblies, revealed an epigenetic landscape in clinical BFG isolates of immense and previously unappreciated diversity. Hundreds of DNA methylation motifs were identified, and most motifs were unique. Though some species (*B. fragilis* and *P. distasonis*) appeared to contain species-specific motifs that could be detected in each analyzed genome, this was uncommon, and almost all motif combinations were observed only in single isolates. Furthermore, DNA methylation motif composition varied dramatically even over short phylogenetic distances between genomes of a species, implying profound epigenetic diversity even among closely related lineages within the BFG.

While substantial diversity of DNA methylation patterns has been observed across species within the bacterial domain of life[43], large surveys of DNA methylation diversity between different species within a genus have previously not been conducted, and closely related species have not been compared in a systematic way. Our study puts forward an extensive analysis of the relationships between the methylome, intra- and inter-species phylogeny, and diversity within the BFG, based on a unique historical collection of BFG clinical isolates. The isolate collection on which the study is based has additional features that add significant value to our dataset. First, whereas many

prior BFG studies have focused on strains collected from the GI microbiome, our set contains primarily clinical BFG isolates cultured from sites of infection, whose genomes and methylomes may facilitate studies of how invasive isolates may differ from commensal GI strains. Second, our set spans four decades and stretches back into the pre- and early antibiotic eras for a number of commonly used agents, allowing examination of how both the resistome and methylomes evolved under the selection of these agents over a period of four decades.

Our findings raise the question of whether BFG species have more diverse epigenomes than other pathogens and commensals that inhabit the human GI microbiome. Answering this question is challenging given the limited amount of available data. Rarefaction analysis demonstrated that the panepigenomes of the BFG species we studied remained open without signs of asymptotic saturation over the sequenced set, implying substantial unsampled diversity. Comparison with similar rarefaction analysis of published data from *C. difficile*[9]. and *B. breve*[41] suggested somewhat less intraspecies DNA methylation motif diversity than the BFG. This analysis however comes with a few important caveats. First, it is possible that underlying host genome diversity in *B. breve* and *C. difficile* studies was lower, which could result in an underestimate of methylation motif diversity. Second, it is possible that the present study used more sensitive methods for the detection of methylation motifs, which would also result in greater apparent diversity. More thorough investigations into other species are needed to establish whether methylome diversity is indeed impacted by phylogeny or lifestyle.

Although we did not examine the transcriptional consequences of methylation in this study, previous works has demonstrated significant regulation of transcription by DNA methylation[42]. It may be reasonable to speculate that the epigenomic diversity we observed may generate proportionate transcriptional diversity within populations with fitness consequences that are operated on by selection. Our finding of apparent genome-wide depletion of DNA methylation motifs within individual bacterial lineages has implications for BFG genome evolution. In these cases, there was typically not a concomitant increase in motifs that would result from mutations of the methylated nucleotide of sufficient magnitude to explain the depletion, suggesting that methylation-induced hypermutation is not solely responsible for epigenome-driven genome change. Selection acting to remove methylated motifs that result in deleterious fitness consequences, on the other hand, may explain these findings, as mutations that eliminate methyltransferase recognition need not be restricted to the methylated nucleotide. Further investigation will be required to understand the underlying mechanisms.

In our dataset, we examined the specific question of whether methylation motifs might be positioned to affect the transcription of AMR genes and influence or control the expression of AMR phenotypes. We found that all classes of AMR genes we examined, including the important *cfiA* gene encoding a beta-lactamase mediating carbapenem resistance, contained methylation motifs both in upstream intergenic regions and in the gene body. Furthermore, the overall epigenomic diversity in motifs among isolates was reflected in the diversity of methylation motifs adjacent to and within AMR gene bodies. Given our findings of extensive potential methylation involving all classes of AMR genes we examined, it may be reasonable to expect that transcription of these genes and the resulting resistance phenotypes will be influenced by which methylases are present and their expression. Epigenome diversity-driven heterogeneity in AMR phenotype may carry benefit to BFG populations, and the purifying selection that often occurs with exposure to antibiotics may select certain epigenomic methylation patterns over others.

Linking DNA methylation motifs to cognate DNA methyltransferases on the basis of genomic analysis alone is challenging. Many of the DNA methyltransferases encoded by genes located in bacterial genomes, especially within mobile genetic elements, are functionally silent in most conditions. It has been suggested that inactivating mutations in DNA methyltransferases, or other genetic switches such as invertible promoters that control methyltransferase expression may be a common evolutionary mechanism used to vary transcriptional programs[44]. In fact, nearly all genomes in our set had a greater number of potential DNA methyltransferase genes than detected methylated DNA motifs, suggesting either the presence of pervasive silent methyltransferases within the BFG, or, alternatively, methyltransferases that are not expressed under standard conditions of growth on rich media. While this may carry interesting evolutionary and functional implications, it introduces additional technical challenges in associating specific methyltransferases to specific motifs. Adding further complication in our data set is the fact that most motifs were detected in only one or a few genomes, precluding a systematic approach to establishing linkages, given the variety of co-occurring silent methylases. Additionally, the Nanodisco method applied here has less than 100% sensitivity, so we expect some methylation motifs went undetected[40].

Of the more than 6000 potential methyltransferase genes we discovered within our genomic dataset, the majority were located in the shell or cloud compartments, often associated with mobile genetic elements. These findings are consistent with those of other studies[35,36] and are also consistent with the assumption that many of the methylases are components of restriction modification or other defense systems. Importantly, we found that approximately 1000 of the identified methyltransferase genes were associated with intact prophages. Network analysis of these prophage genomes revealed a remarkable degree of methyltransferase gene flow among disparate phages with apparent modular swaps of methyltransferases, including of different classes, between phage genomes. These findings suggest a fundamental role for genetic exchange between BFG phages as one of the ultimate sources driving BFG epigenome diversity. Future studies will be needed to examine the exact relationships between phage-phage interactions in the natural GI microbiome context in which they occur and how these interactions may have driven the diversification of the BFG methylome.

## Methods

### Isolate storage, growth, and identification

Historical BFG isolates originally cultured from clinical material between 1973 and 2018 were stored either lyophilized or frozen in skim milk media at the National Institutes of Health Clinical Center Department of Laboratory Medicine (Bethesda, MD). Isolates were de-identified and metadata including year and source/site of culture was maintained. Due to this de-identification, it was not possible to rule out that some isolates in the collection may represent multiple samplings from a single patient. The subset of isolate chosen for sequencing from the larger set were selected to maximize diversity over dates, source, species, and AMR profiles, and this selection likely reduced the inclusion of isolates sampled from single patients. It should be noted that a subset of isolates lacked precise information regarding date and/or source of culture. Selected isolates were recovered and passaged from their original historical stocks to confirm their identity using Bruker Biotyper MALDI-TOF mass spectrometry with manufacturer's database (Supplementary Data 2). All isolates were recovered on BD BBL™ CDC Anaerobe 5% Sheep Blood Agar (BD 221734, Becton, Dickinson and Company, Sparks, MD) or BD BBL™ Brucella Agar supplemented with 5% Sheep's Blood supplemented with hemin and vitamin K1 (BD 297716). Incubation was generally performed for 36–72 h in Mitsubishi Anaero Anaerobic gas chambers with BD BBL™ GasPak $CO_2$ Generators (BD 261205) at 35–37 °C with 6% percent CO2. Isolates were manipulated under ambient aerobic conditions. Confirmed BFG isolates were subsequently reisolated and stored at −80 °C in Cryosavers Skim Milk Media Cryovials (Hardy Diagnostics, Santa Maria, CA) for subsequent culturing and experimentation.

## Anaerobic susceptibility testing by agar dilution

Susceptibility testing was performed using the reference agar dilution method as described in the Clinical and Laboratory Standards Institute (CLSI) guidelines (9th Ed., M11) or the Wadsworth-KTL Anaerobic Bacteriology Manual (6th edition). Briefly, all susceptibility testing media was freshly prepared in 100 mm square grided petri dishes filled to 30 mL and used within a week. For inoculum preparation, isolated colonies recovered from frozen stocks were re-isolated to Brucella agar supplemented with 5% Sheep Blood, hemin and vitamin K1 (BD 297716) to be grown for 40–48 h constituting two serial passages. Selected growth was then suspended in Brucella Broth (B3051, Sigma-Aldrich, St-Louis, MO) a concentration of 0.5 McFarland as measured using either a DEN-1B Densitometer (Grant Instruments, Cambridge, UK) or Microscan Turbidity Meter (Dade Behring (now Siemens) Munich, Germany).

Two microliters of each test isolate ($10^5$ cfu/spot) was then applied to freshly prepared Brucella Agar with supplemented with hemin, vitamin K1 (B2926, Sigma-Aldrich) and 5% Laked Sheep's Blood (Hemostat, Dixon, CA) containing the antibiotic and concentration of choice. Tested antibiotic concentrations were consistent with concentration utilized as clinical breakpoints as determined by CLSI. All plates were inoculated with the following quality control organisms: *E. coli* 25922, *B. thetaiotaomicron* (ATCC 29741) and *B. fragilis* (ATCC 25285). Interpretive criteria were based upon CLSI anaerobe breakpoints as follows (antibiotic followed by S:I:R MICs in μg/ml): Moxifloxacin 2:4:8; Ampicillin 0.5:1:2; Ampicillin/Sulbactam: 8/4:16/8:32/16; Clindamycin: 2:4:8; Metronidazole: 8:16:32; Meropenem: 4:8:16; Piperacillin/tazobactam: 32/4:64/4:128/4; Tetracycline: 4:8:16. Susceptibility determinations were made after approximately 48 h growth.

## DNA extraction and sequencing

Multiple BFG colonies from a single isolate were resuspended in either PBS or sterile water for extraction. Extractions for Illumina sequencing were performed with DNeasy Blood & Tissue (Qiagen, Frederick, MD) and NucliSENS easyMag (bioMerieux, Durham, NC). High molecular weight DNA for long read sequencing was extracted either with the Gentra Puregene Yeast and Bacteria kit (Qiagen) using the Gram-negative protocol or a customized Maxwell HT gDNA Blood kit (Promega Corporation, Madison, WI) protocol on the Kingfisher Flex system (ThermoFisher, North Logan, UT) that involved extracting DNA from a volume of bacteria equivalent of 1/5 of a 10 μl inoculation loop in the PBS suspension and using a 120 μl final elution volume. DNA concentrations were determined using a Qubit 4 fluorometer (ThermoFisher) and purity was assessed for select samples with the Nanodrop One (ThermoFisher).

DNA for Illumina sequencing was prepared with the RipTide High Throughput Rapid Library Prep Kit (IGenomX, Carlsbad, CA). Libraries were sequenced to generate 150 bp PE reads on an Illumina HiSeq 2500 (Illumina, San Diego, CA) at the NIH Intramural Sequencing Centre (NISC) and on an Illumina NextSeq 550 instrument in the NIH Clinical Center. Sequencing data were demultiplexed with fgbio v 0.7.0 as per the iGenomX protocol (http://fulcrumgenomics.github.io/fgbio/) and demultiplexed reads from different lanes were merged. Quality control issues of uncertain origin were encountered with a number of Igenomix RipTide libraries resulting in demultiplexed read files with significant barcode-to-barcode mixing between libraries in a given sequencing run. Strict quality control parameters were used to select a subset of these libraries for polishing of long read assemblies in subsequent steps (see Genome Assembly).

For Oxford Nanopore Technologies (ONT) genome sequencing, genomic libraries were prepared from extracted DNA using the ONT Rapid Barcoding Sequencing Kit (SQK-RBK004) and protocol for the ONT R9.4.1 flow cells (ONT, Oxford, UK). Sequencing was performed with an ONT GridION X5 instrument. For DNA methylation motif identification, paired methylation-free libraries were prepared using the Oxford Nanopore Rapid PCR Barcoding Kit (SQK-RPB004) and protocol (RPB_9059_v1_revL_14Aug2019) and sequenced with ONT R9.4.1 flow cells using the ONT GridION Mk1 instrument. The SQK-RPB004 protocol was modified to use 7.5 ng of input genomic DNA and the PCR step was modified to use 7 min and 30 s for the extension step.

For PacBio genome sequencing, the Pacific Biosciences protocol "Preparing multiplexed microbial SMRTbell libraries for the PacBio Sequel System" was used to create libraries from 3 μg of DNA. Sequencing was performed using a Sequel sequencer (Pacific Biosciences) using version 3 SMRT cells and sequencing reagents with 10-h movies.

## Genome assembly

Bioinformatic analyses were primarily performed on the NIH HPC Cluster Biowulf using installed modules and Conda v. 4.8.3 managed environments. Detailed scripts and instructions are provided through Zenodo (https://zenodo.org/record/7510225) Illumina reads were trimmed with Cutadapt v. 2.6[45] and assembled with SPAdes v. 3.13.1[46]. After contigs under 500 bp were removed, assemblies were checked for genome completeness and contamination with CheckM v 1.0.18[47]. Raw reads from assemblies with greater than 98% completeness and less than 2% contaminations were used for polishing of ONT long-read assemblies with Pilon v 1.23[48].

ONT basecalling was performed with standalone Guppy v. 3.3.3 and 3.4.5 using qcat v.1.0.6 demultiplexing. The ONT GridION MK1 instrument was also used for basecalling and demultiplexing using MinKnow 19.12.6 (Guppy v. 3.2.10+aabd4ec, equivalent to Guppy v. 3.4.5). Filtering, assembly, and polishing were managed with Snakemake v 5.13.0[49]. ONT reads were quality controlled using Filtlong v. 0.2.0 (https://github.com/rrwick/Filtlong) with the settings --min_length 1000 --keep_percent 95. Filtered reads were used for assembly with Flye v. 2.7[50] with the –meta flag enabled for most assemblies, but disabled to optimize a subset of assemblies where numerous spurious contigs were generated. The Flye –asm-coverage flag was also set to 100 to avoid the necessity of down sampling ONT sequencing reads to retain as much coverage as possible for subsequent polishing. Iterative Racon v. 1.14.3[51] polishing was performed four times before Medaka v. 0.12.1 (https://github.com/nanoporetech/medaka) was used for a final error correction step followed by Pilon when short reads were available. Circlator v. 1.5.5 "fixstart" option was used on assemblies to reorient chromosomes to a dnaA start or to orient contigs to the predicted gene nearest to the middle. Medaka polished assemblies were again evaluated with CheckM for completeness and assemblies with greater than 90% completeness and less than 3% contamination were retained for subsequent analysis. rRNA Operons were quantified using Barrnap v0.9 (https://github.com/tseemann/barrnap).

To construct PacBio genomes, demultiplexed PacBio Sequel subreads were assembled with the Hierarchical Genome Assembly Process (HGAP4) pipeline within the PacBio SMRT Link version 6.0.0 package or with Canu (version 1.6 or 1.8)[52]. The assembled contigs were circularized using Circlator 1.5.3[53] and corrected reads generated from HGAP4 or Canu. In some cases, draft contigs were circularized by evaluating contig overlaps using Gepard v1.30[54] and manually joining sequences. The circularized chromosome and plasmid sequences were polished with the PacBio SMRTLink version 6.0.0 resequencing pipeline. The FASTA assembly was annotated using the Prokka (version 1.13) pipeline[55].

## Phylogenetic and antimicrobial resistance gene identification

Multi-locus sequence analysis (MLSA) was performed with long read assemblies from this study and references from the NCBI[56]. Frameshift correction was necessary with ONT generated assemblies to facilitate whole gene retrieval for MLSA. MEGAN v.6.19.2[57] was used on DIAMOND v 0.9.33[58] alignments of ONT assemblies to a reference file of

protein sequences from the same species, as determined by Bruker Biotyper, to output a frameshift corrected fasta file as described earlier[59]. All assemblies and references were annotated with Prokka v. 1.4.6[55] using a custom *Bacteroides* protein database, available through Zenodo (https://zenodo.org/record/7510225). Locus tags that matched reference MLSA scheme gene queries[60] with BLAST v 2.10.0+. BLASTn[61] and BLASTx against the fasta nucleotide/protein files output from Prokka were identified for gene retrieval. Annotations that were still truncated due to frameshift were resolved through manual acquisition of the split annotation and intergenic region identified with Prokka. Genes were retrieved by locus tag and concatenated (16S-*dnaJ*-*gyrB*-*hsp60*-*recA*-*rpoB*) for alignment with MEGA X v.10.1.8 using MUSCLE[62] with default parameters. Columns with less than 75% occupancy were removed using trimAL v. 1.4.rev15[63]. RaxML v. 8.2.12[64] was used to generate a phylogenetic tree using 20 tree searchers with the GTRGAMMA model and tested with 500 bootstraps. The unrooted tree was visualized with ggtree[65].

Mash v. 2.3[66] with a sketch size of 10,000 was used for all-versus-all whole genome comparisons using assemblies, without frameshift correction. 1-Mash distance was used as an average nucleotide identity (ANI) estimate and additional reference assemblies retrieved from NCBI were included in comparisons. The heatmap was generated in using R 4.2.1 with ComplexHeatmap v 2.14.0[67] with dendsort applied to hclust distances calculated using the ward D2 method. Abricate (https://github.com/tseemann/abricate) with 80% minimum coverage and 80% minimum identity was used to query AMR genes against a composite database curated for *Bacteroides*[68–71] (https://github.com/thsyd/bfassembly), and 1911 AMR genes were found. Abricate output table can be found in Supplementary Data 12.

GTDB-Tk v2.0.0 was used with default settings with the r207 reference database to classify all genomes in the set[30]. Agreement between GTDB-Tk and MALDI for species identification (agreement of 360/383 genomes or 94.0%) was based on assuming equivalence of *Bacteroides vulgatus* (former name) and *Phocaeicola vulgatus* (new name). Summary metadata for isolates and GenBank References can be found in Supplementary Data 2.

### Pangenome and accessory region characterization

To correct for assembly errors associated with Nanopore sequencing, Proovframe v0.9.7 (and diamond v2.0.8) (https://github.com/thackl/proovframe), was used to correct indels by aligning polished BFG genomes to Genbank nr database (release 245), and replacing indel regions with Ns to improve ORF contiguity (available through Zenodo as https://zenodo.org/record/7510225). If this indel correction step is not performed, ORF counts can be artificially inflated by split ORFs, and gene family-wise calculations can be affected. Proovframe-corrected genomes were then annotated with Prokka. All genome-based analyses aside from those associated with Fig. 1 and associated supplemental figures were performed using the Proovframe-corrected genomes.

PPanGGOLiN v. 1.1.136[31] was used to generate pangenome graphs and statistics. Genomes were grouped by species (MALDI method), and genomes from the same species were used as input to PPanG-GOLiN with default settings. Using these settings, genes were grouped into families within a threshold of 80% average amino acid identity and 80% alignment fraction. To generate rarefaction curves for each pangenome, PPanGGOLiN gene family matrix tables were input into MicroPan[72] rarefaction module with 50 permutations and MicroPan Heaps module with 100 permutations.

PPanGGOLiN pangenome graph files for each species were used as input to PPanGGOLiN rgp[33] with default settings (minimum length of 3000 nucleotides) to find accessory regions ("regions of genome plasticity") and output these regions as fasta files (Supplementary Data 3). Accessory region sequences were aligned "all-vs-all" using BLASTN with flag " -perc_identity 90". Anicalc from the CheckV

package was used to calculate ANI and AF (Alignment Fraction) of each alignment, and number of alignments for each accessory region sequence with ANI > = 95 and AF > = 85 were counted. Note that accessory region sequences can often consist of multiple mobile genetic elements or genomic islands in tandem, and no attempt was made to separate individual elements within these regions with the exception of bacteriophages.

To find phage defense systems in accessory regions, Padloc v1.0.1 with database v1.1.0 was used with default settings[73]. AMR genes were identified using Abricate as described above. DNA methyltransferase genes were identified using DNA Methylase Finder as described below. Bacteriophages were identified using Cenote-Taker 2 v2.1.3 (https://github.com/mtisza1/Cenote-Taker2) with flags "-p false -db virion --lin_minimum_hallmark_genes 2 --circ_minimum_hallmark_genes 2". Then, CheckV v0.7.0 with database v0.6 was used to find prophage borders and estimate completeness of each phage sequence. To find conjugative machinery genes, ORFs for each accessory region sequence were found and translated with prodigal, using flag "-p meta", then all amino acid sequences were queried against a custom HMM database of conjugative machinery models pulled from PFAM (Supplementary Data 13) using hmmer[74] with flag "-E 1e-8". Hits for two or more genes were required to for a positive identification on a given accessory region sequence.

To find and characterize circular plasmids/episomes, Flye assembly info tables were parsed to extract putative circular sequences less than 1.5 megabases in size. In the data set, some short plasmids/episomes were present at high copy numbers (>50 copies per chromosome), and in some cases these high copy number plasmids/episomes were represented in lower copy number in companion libraries sequenced on the same flow cell. We assessed that this was likely library cross-contamination and to reduce the likelihood of artifactual assignment of plasmids/episomes to the incorrect library, we excluded circular contigs with coverage that was either 80% of the coverage value of the bacterial chromosome or less or if the coverage was less than 30-fold on average across the sequence. This may have resulted in an underestimate of the true number of plasmids/episomes. Furthermore, the Flye assembler occasionally artifactually assembles sequences as concatemers of two or more tandem copies. Each circular sequence was aligned to itself with BLASTN, and, if the total length of the alignment was greater than 140% of the total length of the sequence, the episome was trimmed down to one unit length to eliminate potential artifactual tandem duplications. To determine if the filtered sequences had plasmid-associated genes, each sequence was run through MOBsuite[75] followed by RPS-BLAST against the CDD database[76] with flags " -evalue 1e-2 -seg yes". Hits were then cross-checked against a list of models related to plasmid replicases, relaxases, conjugative machinery, integrases, and transposes (Supplementary Data 14). Also, Abricate was run as described above on each sequence. Plasmids/episomes were clustered into approximate operational taxonomic units (OTUs) using anicalc and aniclust from CheckV with flags "--min_ani 95 --min_tcov 85" (minimum ANI = 95%, minimum AF = 85%). The network graph was visualized in Cytoscape[77].

### DNA methylase finder construction

Identification of DNA methyltransferase genes is difficult for at least three reasons: (1) the sequence space of DNA methyltransferase genes/domains is very large and diverse, (2) some DNA methyltransferase domains have homology to other domains, mostly RNA methyltransferase domains, and (3) many DNA methyltransferase genes have multiple domains (e.g., a DNA methyltransferase domain and a DNA helicase) resulting in potential annotation by the comparator gene only (not the methyltransferase gene) by standard annotation tools. Furthermore, many genes annotated with Prokka or NCBI's Prokaryotic Genome Annotation Pipeline (https://github.com/ncbi/pgap) are labeled as "methylase", and it is unclear if these genes are

DNA methyltransferases, RNA methyltransferases, protein methyltransferases, or something else.

To solve these issues, DNA Methylase Finder was created. A complete description of all components of the pipeline with documentation, as well as the fully executable version used in this work are available at https://github.com/mtisza1/DNA_methylase_finder. At the beginning of the pipeline, input protein sequences (or translated nucleotide inputs) are queried using hmmer against a custom database of HMMs from diverse DNA methyltransferase domains from PFAM, CDD, PDB, the work of Oliveira et al 2014, and additional models generated in-house (https://zenodo.org/record/6647341/)[37]. Aligned proteins are trimmed down to just the aligned region representing the putative DNA methyltransferase domain, and these regions are then queried against all of CDD using hmmer to see if any other models (such as RNA methyltransferase domains) are a better match. If a DNA methyltransferase model remains as the best hit, the putative DNA methyltransferase is typed (i.e., Type I, Type II, Type IIG, Type III) using subtype specific models from Oliveira et al, 2014, and prospective motif specificity is inferred by BLASTP alignment to REBASE database DNA methylases (http://rebase.neb.com/rebase/rebase.html) with known motif specificity (default 80% AAI and 80% AF threshold to report specificity). Finally, if nucleotide contigs/genomes were used as an input, maps of DNA methylase "gene neighborhoods" (flanking +/− 5 genes) are annotated with models to restriction enzymes, specificity subunit genes, followed by all of CDD. This tool and documentation are available on GitHub (https://github.com/mtisza1/DNA_methylase_finder). Databases are available at https://zenodo.org/record/6647341/.

To evaluate the sensitivity of DNA Methylase Finder, the REBASE "gold standard" database of DNA methyltransferases was used (downloaded May 21, 2021). These protein sequences were entered as input to DNA Methylase Finder with default settings. Conversely, to assess a false positive rate, all (6011) of the putative DNA methyltransferase genes sequences from the BFG genomes were extracted and compared, via BLASTP with 1e-3 evalue threshold, to the REBASE "gold standard" database of DNA methyltransferases.

In the sensitivity test, DNA Methylase Finder identified 100% of the microbial methyltransferases in the set. The only putative methyltransferase proteins in this REBASE database that were not identified by DNA Methylase Finder were mouse and human methyltransferase genes as well as two sulfotransferase genes (e.g., M.SenCer87DndC) that may have been added to the database in error.

In the assessment of false positive rates, we found that 329/6011 (5.4%) putative methyltransferase genes identified had no hit to the REBASE database at this e-value cutoff. While a number of motifs in the set of 320 had apparent high quality DNA methyltransferase domains based on manual HHpred searches, others appeared to be true false positives. We thus estimate the false positive rate based on this comparison to be up to 5.4%.

## Phage-encoded DNA methyltransferases

Prophage sequences were extracted as described above, and virus OTUs were generated by clustering using anicalc and aniclust from CheckV with flags "--min_ani 95 --min_tcov 85" (minimum ANI = 95%, minimum AF = 85%). Genome maps were drawn with Cenote-Taker 2, and related genomes were visualized with Clinker v0.0.21[78]. DNA methylase genes were identified with DNA Methylase Finder, and clustered with aniclust based on 80% average amino acid identity and 80% alignment fraction with alignments derived from "all-vs-all" BLASTP search.

## Identifying methylated DNA motifs with Nanodisco

Genomic DNA from 268 BFG isolates from five species (*B. fragilis, B. ovatus, B. vulgatus, B. thetaiotaomicron*, and *P. distasonis*) were prepared with the Oxford Nanopore SQK-RPB004 kit as described above. These data and data from isolate-matched "native" (SQK-

RBK004) genomic DNA sequencing were (re)-base called with Guppy 5.0.7 "hac/high-accuracy" mode. Nanodisco v1.0.3[40] was used per instructions, with 300 "chunks" being analyzed for each genome (Nanodisco difference option). Following data processing with Nanodisco, all genomes received manual motif curation, as a significant proportion of the potential motifs that are given as initial output are likely incorrect (usually too specific or too broad). Expert curation was performed by a single operator (MT) and was based on detailed analysis of the output of the program. Expert curation involved identification and correction of two common types of errors. One error involved merging of similar motifs and a second error involved truncation of motifs. Full explanation of the steps for how these two errors were identified and corrected are shown in examples in Supplementary Figs. 8–9.

Motifs that were ultimately cataloged were required to demonstrate an obvious signal difference at nearly all motif occurrences (according to inspection of "Refine_motifs" plots). We expect that the method we applied to manual curation represents a conservative approach and it is possible that it excluded actual methylation motifs that were poorly detected by Nanodisco. Note that Type I DNA methyltransferases target a gapped motif and its reverse complement (e.g., TCANNNNNGTC/GACNNNNNTGA). For the purpose of analysis, we made the decision to count non-palindromic motifs putatively targeted by the same DNA methyltransferase as separate motifs. We applied the same counting logic to the external data that was used in analyses[9,41].

## Comparing Pacbio and Nanopore/Nanodisco de novo motif detection

For the isolates with genome sequencing data from Pacbio, Nanopore native, and Nanopore PCR, de novo motif calls were performed by both the default Pacbio pipeline and Nanodisco. Output tables were compared (Supplementary Table 2).

## Quantifying motifs in BFG genomes

For each motif, seqkit locate[79] with flags to allow for ambiguous bases, such as N or W was used to scan for all instances of each motif on all relevant genomes. Comparison of motif abundance between lineage ingroups versus outgroups was done in python with stats annotator v0.4.3 package (https://github.com/trevismd/statannotations) using $T$ tests with Benjamini-Hochberg correction (1% FDR).

Similarly, to assess motif density in genes (Supplementary Fig. 13), prokka-output gene sequences were evaluated with seqkit as above. Seqkit was also used to ascertain gene length and GC%. Abricate was run on all genes (as described above) to identify and annotate AMR genes.

## Statistics & reproducibility

As this study involved retrospective sequencing of available stored clinical isolates, no statistical method was used to predetermine sample size. Sequencing libraries demonstrating either evidence of contamination or poor quality were discarded and repeated; no other data were excluded from the analyses. This work involved only sequencing, methylome determination, and bioinformatics analysis of de-identified bacterial isolates and thus no randomization was required or performed. The investigators were not blinded for any of the analyses. De-identified BFG isolates were obtained from a frozen, stored, historical collection, and thus there were no pre-registered criteria for this collection.

## Ethics statement

The work presented in this manuscript involved only de-identified clinical bacterial isolates. As such, this work was excluded from NIH IRB review under OHSRP exemption 19-NIAID-00802.

## Reporting summary

Further information on research design is available in the Nature Portfolio Reporting Summary linked to this article.

## Data availability

The raw sequencing data generated in this study have been deposited in the NCBI database under BioProject accession code PRJNA646575. Supplemental Data File 3, and the raw data for PacBio and nanopore genome construction and nanodisco analysis, and instructions are located in a zenodo database under https://zenodo.org/record/7510225 and https://zenodo.org/record/7548812. The primary FAST5 output files from nanopore sequencing are available on request and have not been uploaded to a public repository due to file size (>10 Tb). Requests for materials associated this work require a standard NIH Material Transfer Agreement with the NIH and U.S. Government. Requests for materials should be addressed to John Dekker at john.dekker@nih.gov.

## Code availability

Methylase Gene Finder is available for the Linux command line through GitHub https://github.com/mtisza1/DNA_methylase_finder, with an associated database deposited at https://zenodo.org/record/6647341/. Other scripts are located within https://zenodo.org/record/7510225.

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

## Acknowledgements

We thank the staff of the Microbiology Service in the Dept. Lab Medicine, NIH Clinical Center for technical support and acknowledge Morgan Park

of NISC for assembly of the PacBio genomes. This work was funded by the Intramural Research Program of the National Institute of Allergy and Infectious Diseases (NIAID) and utilized the computational resources of the NIH HPC Biowulf cluster. (http://hpc.nih.gov). The content and views expressed in this work are those of the authors and do not necessarily represent the official views of the NIH or U.S. Government.

## Author contributions

M.J.T., D.D.N.S., A.E.C., P.P.K., and J.P.D. conceived of and designed the study. J.P.D. obtained and managed funding for the study. A.E.C. curated the isolate collection and managed isolate metadata documentation. A.E.C. and D.D.N.S. performed susceptibility testing. M.J.T., D.D.N.S., A.E.C., and J.-H.Y. conducted Illumina and/or Nanopore genomic sequencing. The NISC Comparative Sequencing Program of NHGRI, NIH conducted Illumina and PacBio sequencing of selected isolates. Morgan Park of NISC performed assembly of microbial genomes from PacBio reads. M.J.T. and J.P.D. planned Nanopore methylome sequencing experiments. M.J.T. conducted Nanopore methylome sequencing and methylome data analysis. M.J.T. created the DNA Methylase Finder tool and performed methyltransferase identification and analysis. M.J.T., P.P.K., and D.D.N.S. performed computational analyses of genomic data and performed critical data management. M.J.T. and D.D.N.S. generated primary manuscript and supplementary figures. J.P.D. supervised the study. M.J.T., D.D.N.S., P.P.K., and J.P.D. conducted critical review of both experimental data and computational analyses. M.J.T., D.D.N.S., A.E.C., and J.P.D. wrote and revised the manuscript. All authors critically reviewed and/or edited the manuscript.

## Funding

## Competing interests

The authors declare no competing interests.

## Additional information

## NISC Comparative Sequencing Program

Beatrice B. Barnabas[3], Sean Black[3], Gerard G. Bouffard[3], Shelise Y. Brooks[3], Juyun Crawford[3], Holly Marfani[3], Lyudmila Dekhtyar[3], Joel Han[3], Shi-Ling Ho[3], Richelle Legaspi[3], Quino L. Maduro[3], Catherine A. Masiello[3], Jennifer C. McDowell[3], Casandra Montemayor[3], James C. Mullikin[3], Morgan Park[3], Karen Schandler[3], Brian Schmidt[3], Christina Sison[3], Sirintorn Stantripop[3], James W. Thomas[3], Pamela J. Thomas[3], Meghana Vemulapalli[3] & Alice C. Young[3]

[3]National Human Genome Research Institute, NIH, Bethesda, MD, USA.

