## [Peer Review File · Nature Communications]

REVIEWER COMMENTS

Reviewer #1 (Remarks to the Author):

Tisza et al. did pangenomic and panepigenomic analysis using more than 300 clinic human GI BFG isolates and using long-read sequencing technologies, mainly Oxford Nanopore sequencing, identifying isolate-specific DNA methylation motifs, characterized DNA methylation motifs pattern in BFG and also reveals some biological imply, especially prophage associations. The workload in this research is huge and data obtained are solid and powerful. The findings are meaningful and significant and but not really exciting. Especially due to the de-identification of the clinic human GI BFG isolates, there is no further deeper analysis of these BFG genetic and epigenetic diversity and differentiation in the role of host-BFG association or any clinical imply. Authors should also evaluate, describe and discuss the accuracy of nanopore sequencing and analysis pipelines for BFG DNA methylation analysis. It is good that authors used a small representative subset (n=13) with additional PacBio SMRT sequencing. confirmation between nanopore and SMRT in Genetic and epigenetic analysis should be in main text.

Reviewer #2 (Remarks to the Author):

Tisza and co-workers build upon third generation sequencing to perform a large-scale genomic and epigenomic analysis of the *Bacteroides fragilis* group. This is a well-written and timely study, for which I have a few comments/suggestions, which I hope will help the authors improve their manuscript.

1) A new tool, MethylFinder, is proposed to map bacterial DNA MTases and their vicinity. I did not find a benchmark between this tool and other available ones such as REBASE or DefenseFinder. A sensitivity/specificity analysis is missing, as we need to know how many false/true positives/negatives are popping out from it.

2) Does the tool allow to detect vicinal REases? This would be important to classify each MTase as solitary or part of an R-M system.

3) The above links with my next question. The authors describe a generally higher number of detected MTases than methylation motifs detected. While the authors' argument regarding the

existence of multiple MTases that are non-functional or not expressed is absolutely plausible (and likely), I am wondering if they are not further inflating the number of MTase hits with a few false positives (for example, coming from highly degraded MTases or other genes sharing similar HMM profiles). Concurring to my doubt, is the authors' finding of a maximum of 38 MTases in a single genome. While I am totally open to novelty, this number surpasses the well-known outlier in bacterial MTase genes *Helicobacter pylori*. A quick look at REBASE pacbio data for a couple of species presented here, shows much lower gene predictions (<10 for *ovatus* for example). I really think the authors should double-check the accuracy of their predictions.

4) Regarding Nanodisco, the authors state in their MM section: "all genomes received manual motif curation, as a significant proportion of the potential motifs that are given as initial output are likely incorrect (usually too specific or too broad). Expert curation was performed by a single operator (MT) and was based on detailed analysis of the output of the program. Motifs that were ultimately cataloged were required to demonstrate an obvious signal difference at nearly all motif occurrences (according to inspection of "Refine_motifs" plots)". Knowing by my own experience how Nanodisco can fail in its predictions, I think it is absolutely crucial (for reproducibility purposes), that the authors clarify what do they mean by "detailed analysis of the output" and "obvious signal difference". Maybe they could present in Sup. Mat. some representative examples of their analysis strategy.

5) Characteristically, type I MTases recognize bipartite DNA motifs comprising two specific sequences separated by a non-specific spacer, and methylate a base in each specific motif, resulting in one methylated base on each strand. However, we now know that many bacterial species harbor a non-canonical subgroup of type I R-M enzymes that possess two different M subunits and catalyze, for example, m6A modification on one strand and m4C modification on the other. Was this possibility taken into consideration?

6) Motif abundance analyses in the gene body and upstream/promoter region of AMR genes were, if I well understood, simply performed via presence/absence using seqtk. I think the latter are missing some stats. One possibility is to compare the observed counts with the expected count under a Markov chain model. Are these motifs enriched in AMR compared with other gene families? If so, is this enrichment at the level of the gene body, or upstream/promoter regions?

7) I understand that this is a broad epigenetic analyses, and that the authors have already stated that the interplay between methylation and AMR warrants further investigation. But it's actually a pity that at least for the core/quasi-core MTases, the authors missed an opportunity to perform a DE analysis between mutant and WT. This would give us important functional clues on the extent that these MTases are impacting the biology of the BFG.

Response to Reviewers

Roving methyltransferases generate a mosaic epigenetic landscape and influence evolution in *Bacteroides fragilis* group

Michael J. Tisza, Derek D. N. Smith, Andrew E. Clark, Jung-Ho Youn, NISC Comparative Sequencing Program, Pavel P. Khil, John P. Dekker

Manuscript NCOMMS-23-03484A

We sincerely thank the two reviewers for the extremely thorough and thoughtful reviews of our manuscript. These critical comments and suggestions helped us to clarify a number of issues and have substantially improved the manuscript, and we are very highly appreciative of the effort and significant time donated to critique our work. Below we answer each of the questions raised. In particular, (1) we have addressed the requested DNA Methylase Finder benchmarking and sensitivity and specificity analyses, (2) we have included additional methods and examples of how the methylation motifs were curated from the raw Nanodisco output, and (3) we have done additional analyses of motif abundance near AMR gene bodies relative to other gene families. We believe that with the analysis and explanations below and associated modifications of the manuscript itself that we have met all of the reviewer's concerns.

Reviewer #1 (Remarks to the Author):

Tisza et al. did pangenomic and panepigenomic analysis using more than 300 clinic human GI BFG isolates and using long-read sequencing technologies, mainly Oxford Nanopore sequencing, identifying isolate-specific DNA methylation motifs, characterized DNA methylation motifs pattern in BFG and also reveals some biological imply, especially prophage associations. The workload in this research is huge and data obtained are solid and powerful. The findings are meaningful and significant and but not really exciting. Especially due to the de-identification of the clinic human GI BFG isolates, there is no further deeper analysis of these BFG genetic and epigenetic diversity and differentiation in the role of host-BFG association or any clinical imply. Authors should also evaluate, describe and discuss the accuracy of nanopore sequencing and analysis pipelines for BFG DNA methylation analysis. It is good that authors used a small representative subset (n=13) with additional PacBio SMRT sequencing. confirmation between nanopore and SMRT in Genetic and epigenetic analysis should be in main text.

Response: We greatly appreciate this reviewer's very careful and thoughtful review of our manuscript and assessment of the data as solid, powerful, and meaningful. In response to the comments of both reviewers, we have done the following:

1. We have further discussed the Nanodisco approach used to identify methylation motifs and the methods that we applied to curate motifs from the raw data output. These efforts are further explained in response to the related question #4 from reviewer 2. In summary,

we have added text to the main manuscript (lines 249-259, 704-404) and provided two examples demonstrating two distinct, but common, errors (motif merging and truncation) returned by Nanodisco, and how these errors were handled in the supplemental information (added as Supplementary Figs. 9-10).

2. We have performed an explicit comparison of the PacBio motif calls and the ONT/Nanodisco motif calls for the six isolates for which we obtained methylation calls by both techniques (added as Supplementary Table 2, included as Table RR1 below, and explained in lines 249-256).

Table RR1

BFG-100 (*B. caccae*)

Motif	Modification	Nanodisco	PacBio
CCATC	6mA	Y	Y
GATGG	6mA	Y	Y
AAGNNNNNTCC	6mA	Y	Y
GGANNNNNNCTT	6mA	Y	Y
GAAGNNNNNNGT	6mA	Y	Y
AACNNNNNNCTTC	6mA	Y	Y

BFG-121 (*B. stercoris*)

Motif	Modification	Nanodisco	PacBio
CCNAG	6mA	Y	Y
GATC	6mA	Y	Y
CTKMAG	6mA	Y	Y
TAARAYC	6mA	Y	Y
CNACNNNNNGGC	6mA	Y	Y
GCCNNNNNGTNG	6mA	Y	Y

BFG-250 (*B. cellulosilyticus*)

Motif	Modification	Nanodisco	PacBio
RGATCY	6mA	Y	Y
AGCAG	6mA	Y	Y
GGTNACC	6mA	Y	N
CAGNNNNNTGG	6mA	Y	N
CCANNNNNNTCTG	6mA	Y	N

BFG-238 (*P. distasonis*)

Motif	Modification	Nanodisco	PacBio
GTANNNNNNGTC	6mA	Y	Y
GACNNNNNNTAC	6mA	Y	Y
CCAGG	6mA	Y	Y
CCTGG	4mC	Y	Y
CTCGAG	6mA	Y	Y
RGATCY	6mA	Y	Y
CGTCGAG	6mA	N	Y
CGCG	5mC	Y	N

BFG-256 (*B. salyersiae*)

Motif	Modification	Nanodisco	PacBio
GATC	6mA	Y	Y
AAGACC	6mA	Y	Y
TCANNNNNGTTY	6mA	Y	Y
RAACNNNNNTGA	6mA	Y	Y
CCANNNNNNNTGG	6mA	Y	Y
GGANGAC	6mA	Y	Y
CTAG	4mC	Y	Y
GGNCC	5mC	Y	N

BFG-1 (*B. fragilis*)

Motif	Modification	Nanodisco	PacBio
CTCAT	6mA	Y	Y
CCAAG	6mA	Y	Y
AGCNNNNRTTG	6mA	Y	Y
CAAYNNNGCT	6mA	Y	Y

- Based on *de novo* detection, the methods were in strong agreement for 6mA (29/33 possible matches) and 4mC modifications (2/2 possible matches). The PacBio method used for this sequencing was non-HiFi sequencing (prior to its widespread adoption), which was reported not to detect 5mC methylation reliably. Consistent with this, the two 5mC motifs detected with Nanopore/Nanodisco were not detected by PacBio.

Reviewer #2 (Remarks to the Author):

Tisza and co-workers build upon third generation sequencing to perform a large-scale genomic and epigenomic analysis of the Bacteroides fragilis group. This is a well-written and timely study, for which I have a few comments/suggestions, which I hope will help the authors improve their manuscript.

Response: We sincerely appreciate this reviewer's highly insightful and greatly detailed review of our manuscript and positive summary of our work above. Below we respond to the comments point-by-point, and the modifications we have made, which significantly improved our manuscript.

1) A new tool, MethylFinder, is proposed to map bacterial DNA MTases and their vicinity. I did not find a benchmark between this tool and other available ones such as REBASE or DefenseFinder. A sensitivity/specificity analysis is missing, as we need to know how many false/true positives/negatives are popping out from it.

Response: We thank Reviewer #2 for this suggestion and agree that the initial version of the manuscript was missing an adequate benchmarking comparison. As with any benchmarking approach, our conclusions will likely be biased by our choices of ground truth data and test data. We surmised that the REBASE database of methyltransferase protein sequences represented the best "ground truth" data for such a sensitivity benchmarking test. The database of protein sequences was thus downloaded and used as input data to DNA Methylase Finder. The result was that DNA Methylase Finder identified 100% of the microbial methyltransferases in the set. The only putative methyltransferase proteins in this REBASE database that were not identified by DNA Methylase Finder were mouse and human methyltransferase genes as well as two sulfotransferase genes (e.g. M.SenCer87DndC) that may have been added to the database by mistake. We believe this is a reasonable and relevant sensitivity test of DNA Methylase Finder, as REBASE is widely regarded as a "gold standard" database.

To test specificity, we again used the REBASE database, and queried the 6,011 putative DNA methyltransferase genes identified by DNA Methylase Finder in the BFG dataset (from this manuscript) against the REBASE database using BLASTP with an e-value cutoff of $1e-3$. This revealed that 329/6011 (5.4%) putative methyltransferase genes identified had no hit to the REBASE database at this e-value cutoff. Of this set of 329, many had unambiguous DNA methyltransferase domains per manual HHpred searches, but others seemed to be likely false positives. Because a more exact number would be difficult to ascertain reliably given the lack of functional characterization of the sequences in question, we believe it is most conservative to report a false positive rate of "up to 5.4%" for DNA Methylase Finder. We have added the

methods used in this sensitivity and specificity benchmarking to the Methods section of the manuscript in lines 670-685 and to the main text in lines 188-190.

2) Does the tool allow to detect vicinal REases? This would be important to classify each MTase as solitary or part of an R-M system.

Response: We thank the reviewer for highlighting this point. Indeed, DNA Methylase Finder annotates proteins upstream and downstream of the identified DNA methyltransferase and reports restriction endonucleases, as well as specificity subunits. To address the reviewer's more specific point, we have collated the outputs from this tool for identified methyltransferases

In Figure RR1A below, we find that specificity subunits are detected almost exclusively near putative Type I DNA methyltransferases, as expected. Figure RR1B demonstrates that restriction endonucleases are detected in the neighborhood of 100% of putative Type III DNA methyltransferases, and most Type II DNA methyltransferases are "orphans", without vicinal restriction endonucleases identified. We have added Figure RR1 to the manuscript as Supplementary Figure 7, as well as additional explanatory text in the main manuscript in lines 218-224.

Figure RR1

A

B

Figure RR1. DNA methyltransferase neighborhood analysis. (A) Summary of specificity subunits in the neighborhood of different putative subtypes of DNA methyltransferases identified in the BFG set. (B) Summary of restriction endonucleases in the neighborhood of different putative subtypes of DNA methyltransferases identified in the BFG set.

3) The above links with my next question. The authors describe a generally higher number of detected MTases than methylation motifs detected. While the authors' argument regarding the existence of multiple MTases that are non-functional or not expressed is absolutely plausible (and likely), I am wondering if they are not further inflating the number of MTase hits with a few false positives (for example, coming from highly degraded MTases or other genes sharing similar HMM profiles). Concurring to my doubt, is the authors' finding of a maximum of 38 MTases in a single genome. While I am totally open to novelty, this number surpasses the well-known outlier in bacterial MTase genes *Helicobacter pylori*. A quick look at REBASE pacbio data for a couple of species presented here, shows much lower gene predictions (<10 for *ovatus* for example). I really think the authors should double-check the accuracy of their predictions.

Response: We thank the reviewer for this detailed thought on our results and these highly salient and important comments. We believe there are a variety of factors at play as to why the number of DNA methyltransferases our pipeline detects is higher than previous reports, which we have explained in detail below, now included in summary form in the text in lines 202-217 with Supplementary Data File 7 and Supplementary Figure 6 as noted.

1. We fully expect our pipeline to have false positives, though, as addressed in the response to question #1, we believe the proportion is probably a relatively small fraction, on the order of not more than ~5% as we identified in our specificity analysis.
2. Regarding the possibility of the presence of non-methyltransferase genes with domains with similarity to methyltransferases our dataset: Each prospective DNA methyltransferase domain considered by the pipeline is queried against all of CDD using hmmer. If the prospective domain has a stronger hit (e-value) to any non-DNA methyltransferase domain (e.g. tRNA methyltransferase), this prospective hit will be discarded. We believe that this approach is relatively stringent and contributed to the relatively low false positive rate found in our specificity analysis. Of course, the ability of this approach to discriminate DNA methyltransferases from non-DNA methyltransferases depends on the quality of input data annotation for the relative comparisons.
3. BFG-632 is the genome with 38 putative DNA methyltransferase genes. We've now provided a separate .fasta file with this genome's DNA methyltransferase gene protein sequences (Supplementary Data File 7).
 - a. First, we note that many of the putative DNA methyltransferase genes have identical/near identical sequences within this genome (i.e. they are multicopy genes). Thus, the large number may be accounted for, in part, by duplications due to transposon insertions. We surmise that the transposons that often have many, non-tandem insertions across a single genome often carry DNA methyltransferases, similar to our observations in Fig. S2B and Fig. 2C. It is possible that short read genome assemblies may underestimate the number of such methyltransferase duplicates due to multiple insertions. This could be examined in a systematic way from public sequencing data (e.g. evaluating for anomalous coverage associated with methyltransferase genes in short read assemblies), but this sort of analysis of public data would probably be beyond the scope of this manuscript.
 - b. We next performed a web-based BLASTP search of the 38 identified proteins using the clustered-NR database. Almost all sequences have top hits over 90% AAI to DNA methyltransferases (annotated by GenBank's pipeline) from other bacteroidaceae genomes. There is one exception for these putative DNA methyltransferases from BFG-632 wherein a gene has a top hit to DUF1156 proteins. However, querying that protein in HHPRED suggests that it has a high confidence methyltransferase domain. These results can be verified independently by the reviewers. We note of course, as above, that the comparative approach used here depends on the accuracy of annotation of the comparison dataset. To

determine the likelihood of contamination in this genome, an important consideration, BFG-632 was run through CheckM, returning a completeness value of 99.08 and a contamination value of 0.16. The raw Flye assembly statistics have no obvious abnormalities, with 7 total contigs, some of which appear to be plasmids.

- c. In the course of our analysis, we noted that BFG species with larger median genome size don't necessarily have higher median DNA methyltransferase gene counts. However, we observed that within each species, there is a positive correlation between genome size and number of putative DNA methyltransferases (illustrated in Figure RR2 and RR3 below, now included as Supplementary Figures 6A and 6B). BFG-632 is the longest genome in the entire collection, so the fact that it has the greatest number of identified DNA methyltransferases fits this overall trend.
4. We believe the REBASE database and REBASE DNA methyltransferase detection strategy are likely quite conservative and confident calls are made only on genes quite similar to validated DNA methyltransferase genes which may be biased towards easily tractable model organisms. In general, a more comprehensive approach may be expected yield more true positive hits than a very conservative approach (in addition to potentially more false positive hits). It's also unclear what proportion of DNA methyltransferases do not engage in motif-driven methylation but are instead involved in DNA repair or other cellular processes, that may be identified by the DNA Methylase Finder approach.

Figure RR2. Distribution of putative methyltransferase genes identified by DNA Methylase Finder for each species.

Figure RR3: Intra-species genome length corresponds to DNA methyltransferase gene number.

4) Regarding Nanodisco, the authors state in their MM section: “all genomes received manual motif curation, as a significant proportion of the potential motifs that are given as initial output are likely incorrect (usually too specific or too broad). Expert curation was performed by a single operator (MT) and was based on detailed analysis of the output of the program. Motifs that were ultimately cataloged were required to demonstrate an obvious signal difference at nearly all motif occurrences (according to inspection of “Refine_motifs” plots).”. Knowing by my own experience how Nanodisco can fail in its predictions, I think it is absolutely crucial (for reproducibility purposes), that the authors clarify what do they mean by “detailed analysis of the output” and “obvious signal difference”. Maybe they could present in Sup. Mat. some representative examples of their analysis strategy.

Response: We agree that the methods applied for manual curation are crucial to interpretation and reproducibility of data. To address this, we’ve provided two distinct examples for how the two most common errors returned by *Nanodisco motif* are identified and corrected, along with explanatory text in lines 224-234 and 664-667. The first example illustrates how two overlapping or locally similar motifs can be “merged” by Nanodisco, how this error can be identified, and how the individual motifs can be resolved, reproduced below as Figure RR4 and added to the manuscript as Supplemental Figure 9. The second example illustrates an example where Nanodisco returns a motif that is “too short” lacking the final discriminatory nucleotide at the end of the motif, how this error can be identified, and how the motif can be corrected, reproduced below as Figure RR5 and added as Supplemental Figure 10.

Figure RR4

Example 1: *B. fragilis* isolate BFG-420 in which **CTCAT** and **CCAGT** motifs are merged.

Step 1: Inspect motif chart from *Nanodisco motif* command, **CYSAK**.

- Note that there are many dots at the baseline in every position.
- This is not consistent with the expectation that the majority of motifs should contain the methylated base.
- A pattern with moderate peaks at '7' and '9' and higher peak at '8' becomes apparent for **CTCAT** motif.

Step 2: Now manually try this motif identified by manual inspection. In this case, **CTCAT**.

- This motif has substantial signal difference at the majority of motif loci. Thus, it is a high-confidence motif.
- But what about that other signal pattern buried within **CYSAK**?

Step 3: Run the *Nanodisco motif* command again to iterate over the data while ignoring **CTCAT**.

- The motif **CCAGT** has been returned, which has the second peak pattern that was visible in the original **CYSAK**.
- We can conclude that *Nanodisco motif* incorrectly merged these motifs based on the "CA" in the middle of each motif.
- The two higher confidence motifs are **CCAGT** and **CTCAT**

Figure RR5

Example 2: *B. thetaiotaomicron* BFG-484 motif missing initial discriminatory G nucleotide.

Step 1: Inspect motif chart from *Nanodisco motif* command, **GCANNNNNNRRTTT**.

- There are many dots that are at the baseline in every position.
- However, there are not 2 distinct signals that emerge.
- This is how a *Nanodisco motif* is returned that is “too short” and missing a discriminatory end nucleotide.

Step 2: Extend the motif with **N**'s on either end, i.e. **NGCANNNNNNRRTTN**

- It now becomes obvious that the "G" added to the beginning of the motif results in high quality identification at this position. Note that the nucleotides at the end of the motif are all equal.

Step 3: Try the original motif plus **G** at the beginning, i.e. **GGCANNNNNNRRTTT**

- The many dots at baseline seen in the original output at the top have now disappeared, and we have identified a motif with much higher confidence.
- Note that it may still seem a little noisy, but if you look at the 5th position, a few distinct, but fully penetrant patterns are clear.

5) Characteristically, type I MTases recognize bipartite DNA motifs comprising two specific sequences separated by a non-specific spacer, and methylate a base in each specific motif, resulting in one methylated base on each strand. However, we now know that many bacterial species harbor a non-canonical subgroup of type I R-M enzymes that possess two different M subunits and catalyze, for example, m6A modification on one strand and m4C modification on the other. Was this possibility taken into consideration?

Response: We thank the reviewer for raising this very interesting point. Indeed, we treated each end of the bipartite motif separately, and we observed this class of Type I motif a number of times (all listed in Supplementary Data 11). For example, *B. fragilis* genome BFG-312 has the motif (RAACNNNNNGGC/ GCNNNNNGTTY) where the underlined "C" was predicted to be 4mC.

6) Motif abundance analyses in the gene body and upstream/promoter region of AMR genes were, if I well understood, simply performed via presence/absence using seqtk. I think the latter are missing some stats. One possibility is to compare the observed counts with the expected count under a Markov chain model. Are these motifs enriched in AMR compared with other gene families? If so, is this enrichment at the level of the gene body, or upstream/promoter regions?

Response: This is a very interesting and important point, and we recognize that we neglected to include a rigorous comparison of motif density in AMR gene bodies vs. the rest of the genome in the manuscript. We have now performed an extensive analysis of methylation in AMR genes relative to non-AMR genes in the five principal species for which we had sufficient numbers of isolates, and summarize the findings below. We found a direct correlation between motif content (motifs per Kb) and GC content. This appeared to be a robust finding in itself, and we did not explore the reasons for this further in this context. Because of this correlation, we thought the best way to analyze the relative motif content of AMR genes vs. all genes is to plot motifs/kb vs. GC%, as we have done in the analysis represented in Figure RR6. In this figure, the AMR genes are plotted as colored dots and non-AMR genes are plotted in black. This analysis demonstrates that there is no clear general enrichment or depletion of methylation motifs in AMR genes compared to non-AMR genes (labeled "NA") across species after controlling for GC%. There may be some depletion in AMR genes of *P. distasonis*, but given the overall findings, we did not explore this specific species further. It can be seen that the AMR genes fall mostly along the linear fit of the data, and this is apparent as well in the Z-scores calculated for the same data to the right. The main conclusion is that there is no obvious or strong statistical difference in motif density (adjusted for CG content) seen in AMR genes vs. the rest of the genome. We have now included this as Supplemental Figure 13 in the manuscript (Figure RR6 below) and noted the results in the text in lines 300-309.

Figure RR6

Figure RR6: Motif density in AMR genes compared to non-AMR genes. All genes from all isolate genomes for which methylation data was available were extracted and methylation motifs found in one or more genomes of each species were mapped to corresponding genes. Antimicrobial resistance genes were called using abricate and the motif density of each gene was normalized by GC%. Z-score was calculated and data were plotted by AMR gene category (non-AMR genes = “NA”). Lines represent iteratively reweighted least squares linear fits of all data in each plot.

7) I understand that this is a broad epigenetic analyses, and that the authors have already stated that the interplay between methylation and AMR warrants further investigation. But it's actually a pity that at least for the core/quasi-core MTases, the authors missed an opportunity to perform a DE analysis between mutant and WT. This would give us important functional clues on the extent that these MTases are impacting the biology of the BFG.

Response: We certainly agree that a deeper transcriptional/differential expression analysis (coupled with functional MIC readout) will add important functional and mechanistic data to our understanding of how methylation may regulate the function of AMR genes and influence resistance. We would note that Oliveira and Fang (PMID: 32417228) highlight how studies of the genetics of methyltransferases require incredibly careful work, including a variety of genetic controls on the methyltransferase gene as well as putative target sites. We would like to do this work by engineering isogenic isolates for optimal comparisons, but there are many challenges associated with genetic engineering of wild type isolates that will need to be solved. We are presently planning pursue these studies in the future, but with current staffing this is unfortunately well beyond the scope of the present study and manuscript.

REVIEWERS' COMMENTS

Reviewer #1 (Remarks to the Author):

This revised version has been much improved. The authors have addresses the reviewers' concerns well. They have made appropriate modifications to the manuscript and provided additional information to strengthen their findings in text. I recommend it for publication.

Reviewer #2 (Remarks to the Author):

The Authors have positively addressed my comments and criticisms.